# MMVAE+: Enhancing the Generative Quality of Multimodal VAEs without Compromises

**Emanuele Palumbo[1,2], Imant Daunhawer[2] & Julia E. Vogt[2]**
[1] ETH AI Center, Zürich, Switzerland
[2] Department of Computer Science, ETH Zürich, Switzerland
`emanuele.palumbo@ai.ethz.ch`

## Abstract

Multimodal VAEs have recently gained attention as efficient models for weakly-supervised generative learning with multiple modalities. However, all existing variants of multimodal VAEs are affected by a non-trivial trade-off between generative quality and generative coherence. In particular mixture-based models achieve good coherence only at the expense of sample diversity and a resulting lack of generative quality. We present a novel variant of the mixture-of-experts multimodal variational autoencoder that improves its generative quality, while maintaining high semantic coherence. We model shared and modality-specific information in separate latent subspaces, proposing an objective that overcomes certain dependencies on hyperparameters that arise for existing approaches with the same latent space structure. Compared to these existing approaches, we show increased robustness with respect to changes in the design of the latent space, in terms of the capacity allocated to modality-specific subspaces. We show that our model achieves both good generative coherence and high generative quality in challenging experiments, including more complex multimodal datasets than those used in previous works.

## 1 Introduction

Multimodal VAEs are a promising class of models for weakly-supervised generative learning. Different from initially proposed models in this class (Suzuki et al., 2016; Vedantam et al., 2018), more recent approaches (Wu & Goodman, 2018; Shi et al., 2019; 2021; Sutter et al., 2020; 2021) can efficiently scale to a large number of modalities. These methodological advances enabled applications in multi-omics data integration (Lee & van der Schaar, 2021; Minoura et al., 2021) and tumor segmentation from multiple image modalities (Dorent et al., 2019).

Several variants of scalable multimodal VAEs have been proposed (Wu & Goodman, 2018; Shi et al., 2019; 2021; Sutter et al., 2021) and their performance is measured in terms of generative quality and generative coherence. While generative quality measures how well a model approximates the data distribution, generative coherence measures the semantic coherence of generated samples across modalities (e.g., see Shi et al., 2019). High generative quality requires generated samples being similar to the test data, while high generative coherence requires generated samples to agree in their semantic content across modalities. For instance, in a dataset of image/caption pairs, conditional generation from the text modality should produce images where the depicted object matches the description in the given caption (e.g. matching color). Ideally, an effective multimodal generative model should fulfill both of these performance aspects. Still, recent work (Daunhawer et al., 2022) shows that the predominant approaches exhibit a non-trivial trade-off between the two criteria, which limits their utility for complex real-world applications.

In this work we focus on mixture-based multimodal VAEs, which show high generative coherence only at the expense of a lack of generative quality, a fact that undermines their performance in realistic settings. Results for models in this class are promising for capturing *shared* information, i.e. information that is communal across modalities on the underlying concept being described, while exhibit a lack of modelling of *private* variation, i.e. modality-specific information for single modalities (see Shi et al., 2019; Sutter et al., 2021; Daunhawer et al., 2022). In an attempt to enhance modelling

of private information, recent work (Sutter et al., 2020) has suggested introducing modality-specific latent spaces in addition to a shared subspace for mixture-based multimodal VAEs. However, we revise such proposed extension and find that an improvement in terms of generative quality comes again at the expense of reduced generative coherence. Most importantly we uncover a relevant shortcoming of existing approaches with separate subspaces, in that we find generative coherence to be overly sensitive to hyperparameters controlling the capacity of private latent subspaces, which in practice calls for expensive model selection procedures to achieve adequate performance.

Incorporating the idea of modelling the latent space as a combination of shared and modality-specific encodings, we propose the MMVAE+, a variant of the mixture-of-experts multimodal VAE (MM-VAE,Shi et al. (2019)) with a novel ELBO, that significantly improves the diversity of the generated samples, without sacrificing semantic coherence. Compared to previously proposed models, our method achieves *both* convincing generative quality *and* generative coherence (Section 4.1). Notably, its performance in terms of both criteria is robust with respect to hyperparameters controlling latent dimensionality, compared to previous methods with separate shared and private latent subspaces. (Section 4.2) Finally, we show our proposed model can successfully tackle a challenging multimodal dataset of image and text pairs (Section 4.3), that was shown to be too complex for existing multimodal VAEs (Daunhawer et al., 2022).

## 2  RELATED WORK

Multimodal generative models are promising approaches for learning from co-occurring data sources without explicit supervision, by exploiting the pairing between modalities as a form of weak-supervision. Previous work has achieved outstanding results for multimodal generative tasks such as image-to-image translation (Zhu et al., 2017; Choi et al., 2018) or text-to-image synthesis (Reed et al., 2016). While these models are designed for specialized tasks limited to a fixed number of modalities, in a lot of real-world settings there is a need for general methods that can leverage large datasets of many heterogeneous modalities. A prominent example is the field of healthcare, where personalised medicine requires learning from large-scale multimodal datasets comprised of medical images, genomics tests, and clinical measurements. Multimodal VAEs (Suzuki et al., 2016; Wu & Goodman, 2018; Shi et al., 2019; 2021; Sutter et al., 2020; 2021) are a promising model class for such applications, showing encouraging results towards efficient learning from multimodal datasets with multiple heterogeneous modalities.

Multimodal VAEs extend the popular variational autoencoder (VAE, Kingma & Welling, 2014) to multiple data modalities. Initially proposed approaches (Suzuki et al., 2016; Vedantam et al., 2018) lack scalability in the number of modalities, requiring an additional encoder network per possible subset of modalities, to enable inference from the given subset. Other proposed methods require explicit supervision (Tsai et al., 2019), which demands prior expensive data labelling processes. In contrast, recent methodological advances enable learning from a large number of modalities efficiently and without explicit supervision, by using a joint encoder that decomposes in terms of unimodal encoders. Previous work proposed three different formulations for the joint encoder: the product-of-experts (MVAE, Wu & Goodman 2018), mixture-of-experts (MMVAE, Shi et al. 2019), and mixture-of-product-of-experts (MoPoE-VAE, Sutter et al. 2021).

Recent work (Daunhawer et al., 2022) shows that existing multimodal VAEs exhibit a tradeoff between two desired performance criteria for multimodal generation, namely generative quality and generative coherence. While generative quality assesses the generative performance of the model for each modality, generative coherence (Shi et al., 2019) examines the learning of shared information by estimating the consistency in the semantic content between modalities in both conditional and unconditional generation. Daunhawer et al. (2022) show that existing product-based models exhibit low generative coherence, while mixture-based models exhibit a lack of sample diversity, which negatively affects generative quality (cp. Wolff et al., 2021).

Based on the three mentioned formulations of multimodal VAEs, subsequent work introduced additional regularization terms (Sutter et al., 2020; Hwang et al., 2021) and hierarchical latent spaces (Sutter & Vogt, 2021; Vasco et al., 2022; Wolff et al., 2022). Previous work has also explored the possibility of assuming separate modality-specific latent subspaces in addition to a shared subspace (Sutter et al., 2020; Lee & Pavlovic, 2021; Wang et al., 2016), or leveraging mutual supervision (Joy et al., 2022). Yet, it is not clear whether these extensions overcome the fundamental tradeoff be-

tween generative quality and coherence. For instance, for existing approaches to employ additional modality-specific latent subspaces with mixture-based models, we observe an overly high sensitivity to hyperparameters (Section 4.2), which raises fundamental questions with regard to model selection. Notably, we show that our proposed method—also in the class of multimodal VAEs with modality-specific latent subspaces—is significantly more robust to changes in the same hyperparameter values.

## 3 METHOD

### 3.1 PRELIMINARIES: MMVAE

Given a set of $M$ modalities $\boldsymbol{X} := \boldsymbol{x}_1, \ldots, \boldsymbol{x}_M$, multimodal VAEs learn a latent-variable generative model of the form $p_\Theta(\boldsymbol{X}, \boldsymbol{z}) = p(\boldsymbol{z}) \prod_{m=1}^{M} p_{\theta_m}(\boldsymbol{x}_m | \boldsymbol{z})$, by using neural networks to approximate the intractable evidence $p_\Theta(\boldsymbol{X})$. Analogous to the VAE (Kingma & Welling, 2014), multimodal VAEs optimise a lower bound of the log-evidence by maximizing the following objective:

$$\mathcal{L}_{\text{VAE}}(\boldsymbol{x}_{1:M}) = \mathbb{E}_{q_\Phi(\boldsymbol{z}|\boldsymbol{X})}\left[\log \frac{p_\Theta(\boldsymbol{X}, \boldsymbol{z})}{q_\Phi(\boldsymbol{z}|\boldsymbol{X})}\right] \tag{1}$$

where $q_\Phi(\boldsymbol{z}|\boldsymbol{X})$ denotes the approximate posterior, a neural network that is parameterized by $\Phi$. In the above form, the objective does not scale to a large number of modalities (e.g., see Wu & Goodman, 2018; Sutter et al., 2021) and additional assumptions are required for the joint encoder to be computed efficiently in terms of the unimodal encoders. Different formulations of the joint encoder have been proposed (Wu & Goodman, 2018; Shi et al., 2019; Sutter et al., 2021), but here we focus on the decomposition used by the MMVAE (Shi et al., 2019), formulating the joint encoder as a mixture-of-experts of unimodal encoders $q_\Phi(\boldsymbol{z}|\boldsymbol{X}) = \frac{1}{M} \sum_{m=1}^{M} q_{\phi_m}(\boldsymbol{z}|\boldsymbol{x}_m)$. When a mixture distribution for the joint encoder is assumed, the expression in eqn. (1) is evaluated by computing the expectation with respect to each component of the mixture distribution, and averaging over the results. As in this case we assume a mixture-of-experts encoder, the MMVAE (Shi et al., 2019) objective is a sum indexed by the $M$ unimodal encoders

$$\mathcal{L}_{\text{MMVAE}}(\boldsymbol{x}_{1:M}) = \frac{1}{M} \sum_{m=1}^{M} \mathbb{E}_{q_{\phi_m}(\boldsymbol{z}|\boldsymbol{x}_m)}\left[\log \frac{p_\Theta(\boldsymbol{X}, \boldsymbol{z})}{q_\Phi(\boldsymbol{z}|\boldsymbol{X})}\right] \tag{2}$$

where

$$\mathbb{E}_{q_{\phi_m}(\boldsymbol{z}|\boldsymbol{x}_m)}\left[\log \frac{p_\Theta(\boldsymbol{X}, \boldsymbol{z})}{q_\Phi(\boldsymbol{z}|\boldsymbol{X})}\right] = \mathbb{E}_{q_{\phi_m}(\boldsymbol{z}|\boldsymbol{x}_m)}\left[\log p_{\theta_m}(\boldsymbol{x}_m | \boldsymbol{z})\right] + \sum_{\substack{n=1 \\ n \neq m}}^{M} \mathbb{E}_{q_{\phi_m}(\boldsymbol{z}|\boldsymbol{x}_m)}\left[\log p_{\theta_n}(\boldsymbol{x}_n | \boldsymbol{z})\right] +$$

$$\mathbb{E}_{q_{\phi_m}(\boldsymbol{z}|\boldsymbol{x}_m)} \log\left[\frac{p(\boldsymbol{z})}{q_\Phi(\boldsymbol{z}|\boldsymbol{X})}\right] \tag{3}$$

For each term in the sum, the encoding $\boldsymbol{z} \sim q_{\phi_m}(\boldsymbol{z}|\boldsymbol{x}_m)$ is sampled from a unimodal encoder, but it is used for the reconstruction of *all* modalities: the likelihood term encompasses the reconstruction of the modality $m$, i.e. self-reconstruction $\mathbb{E}_{q_{\phi_m}(\boldsymbol{z}|\boldsymbol{x}_m)}[\log p_{\theta_m}(\boldsymbol{x}_m|\boldsymbol{z})]$, as well as the reconstruction *across* modalities, i.e. cross-modal reconstruction $\sum_{m=1, n \neq m}^{M} \mathbb{E}_{q_{\phi_m}(\boldsymbol{z}|\boldsymbol{x}_m)}[\log p_{\theta_n}(\boldsymbol{x}_n|\boldsymbol{z})]$. Maximizing reconstruction likelihood of modalities not observed for inference encourages the latent code to contain information common across modalities, and can explain improved semantic coherence compared to previously proposed product-based approaches (see qualitative results in Shi et al. (2019) and section 4.1). However it has also been recently been shown that the optimization of cross-modal reconstructions given a unimodal encoding can lead to a lack of modelled diversity in private information for generation across modalities (Wolff et al., 2021; Daunhawer et al., 2022), which manifests in average-looking samples with low generative quality (see section 4.1).

### 3.2 PRELIMINARIES: ADDITIONAL MODALITY-SPECIFIC SUBSPACES

In an attempt to enhance modelling of private information and improve performance, recent work on multimodal VAEs (Sutter et al. (2020)) has proposed to model the latent space for mixture-based

models including modality-specific latent subspaces, in addition to a shared subspace (Bouchacourt et al., 2018; Tsai et al., 2019; Wang et al., 2016) . Formally, each modality $\boldsymbol{x}_m$ is modelled to have its own modality-specific latent code $\boldsymbol{w}_m$, in addition to a latent code $\boldsymbol{z}$ shared between all modalities. The assumed generative model is then of the form $p_\Theta(\boldsymbol{X}, \boldsymbol{z}, \boldsymbol{W}) = p(\boldsymbol{z}) \prod_{m=1}^M p_{\theta_m}(\boldsymbol{x}_m | \boldsymbol{z}, \boldsymbol{w}_m) p(\boldsymbol{w}_m)$, where the shared latent code $\boldsymbol{z}$ and all the modality-specific codes $\boldsymbol{w}_1, \dots \boldsymbol{w}_M =: \boldsymbol{W}$ are assumed to be independent. A variational encoder $q_\Phi(\boldsymbol{z}, \boldsymbol{W} | \boldsymbol{X})$ is introduced to approximate posterior inference, and is modelled in line with the assumptions on the generative process, assuming conditional independence given $\boldsymbol{X}$

$$q_\Phi(\boldsymbol{z}, \boldsymbol{W} | \boldsymbol{X}) = q_{\Phi_z}(\boldsymbol{z} | \boldsymbol{X}) q_{\Phi_{\boldsymbol{W}}}(\boldsymbol{W} | \boldsymbol{X}) = q_{\Phi_z}(\boldsymbol{z} | \boldsymbol{X}) \prod_{m=1}^M q_{\phi_{\boldsymbol{w}_m}}(\boldsymbol{w}_m | \boldsymbol{x}_m)$$

Again, achieving scalability in the number of modalities requires additional assumptions for the joint encoder $q_{\Phi_z}(\boldsymbol{z} | \boldsymbol{X})$ of the shared latent code $\boldsymbol{z}$. In particular assuming a mixture-of-experts joint encoder $q_{\Phi_z}(\boldsymbol{z} | \boldsymbol{X}) = \frac{1}{M} \sum_{m=1}^M q_{\phi_{z_m}}(\boldsymbol{z} | \boldsymbol{x}_m)$ one obtains a variant of the MMVAE with factorized latent space into shared and private subspaces

$$\mathcal{L}_{\text{MMVAE}}^f(\boldsymbol{x}_{1:M}) = \frac{1}{M} \sum_{m=1}^M \mathbb{E}_{\substack{q_{\phi_{z_m}}(\boldsymbol{z}|\boldsymbol{x}_m) \\ q_{\Phi_{\boldsymbol{W}}}(\boldsymbol{W}|\boldsymbol{X})}} \left[ \log \frac{p_\Theta(\boldsymbol{X}, \boldsymbol{z}, \boldsymbol{W})}{q_{\Phi_z}(\boldsymbol{z}|\boldsymbol{X}) q_{\Phi_{\boldsymbol{W}}}(\boldsymbol{W}|\boldsymbol{X})} \right] \quad (4)$$

where

$$\mathbb{E}_{\substack{q_{\phi_{z_m}}(\boldsymbol{z}|\boldsymbol{x}_m) \\ q_{\Phi_{\boldsymbol{W}}}(\boldsymbol{W}|\boldsymbol{X})}} \left[ \log \frac{p_\Theta(\boldsymbol{X}, \boldsymbol{z}, \boldsymbol{W})}{q_{\Phi_z}(\boldsymbol{z}|\boldsymbol{X}) q_{\Phi_{\boldsymbol{W}}}(\boldsymbol{W}|\boldsymbol{X})} \right] = \mathbb{E}_{\substack{q_{\phi_{z_m}}(\boldsymbol{z}|\boldsymbol{x}_m) \\ q_{\phi_{\boldsymbol{w}_m}}(\boldsymbol{w}_m|\boldsymbol{x}_m)}} \left[ \log p_{\theta_m}(\boldsymbol{x}_m | \boldsymbol{z}, \boldsymbol{w}_m) \right] +$$

$$\sum_{\substack{n=1 \\ n \neq m}}^M \mathbb{E}_{\substack{q_{\phi_{z_m}}(\boldsymbol{z}|\boldsymbol{x}_m) \\ q_{\phi_{\boldsymbol{w}_n}}(\boldsymbol{w}_n|\boldsymbol{x}_n)}} \left[ \log p_{\theta_n}(\boldsymbol{x}_n | \boldsymbol{z}, \boldsymbol{w}_n) \right] +$$

$$\mathbb{E}_{\substack{q_{\phi_{z_m}}(\boldsymbol{z}|\boldsymbol{x}_m) \\ q_{\Phi_{\boldsymbol{W}}}(\boldsymbol{W}|\boldsymbol{X})}} \left[ \log \frac{1}{q_{\Phi_z}(\boldsymbol{z}|\boldsymbol{X}) q_{\Phi_{\boldsymbol{W}}}(\boldsymbol{W}|\boldsymbol{X})} \right]$$

Existing mixture-based multimodal VAEs with a factorized latent space are based on this objective Sutter et al. (2020). Note that both self-reconstruction likelihood and cross-modal reconstruction likelihood terms are computed given the modality-specific encoding from the *target* modality. This can seemingly be an advantage, given these additional latent encodings can provide the modality-specific information for reconstruction that is not modelled in the shared latent code. In particular, for cross-modal reconstruction no private information about the target modality can be present in the $\boldsymbol{z}$ encoding from the starting modality.

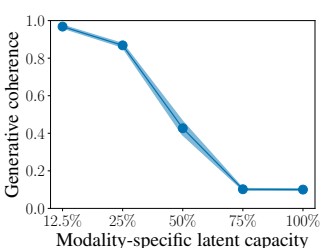

Figure 1: Shortcut problem for the objective in 4: generative coherence as a function of of private latent capacity. The percentages on the x-axis indicate the relative size of any private subspace with respect to the shared subspace.

However, we find that with such an approach model performance is heavily dependent on hyperparameters controlling private latent dimensionality. In fact, when modality-specific latent subspaces are given enough capacity, in terms of number of dimensions, the modality-specific encoding can contain *all* information for the given modality (i.e., both shared and modality-specific features), while the shared subspace is ignored by the decoders. As it was already observed in Wang et al. (2016) in a similar setting, we show here that this approach can lead to a degenerate solution— a *shortcut* where all information flows through the modality-specific subspaces and no shared subspace is learned for generation across modalities. As a consequence, to achieve sufficient generative coherence it is necessary to heavily constrain the size of private subspaces (see figure 1). Note however that this in turn negatively impacts generative quality, being detrimental to modelling modality-specific variation, which was our motivation to employ additional private subspaces in the first place (see section 4.2). Therefore, assuming modality-specific latent subspaces seems to not be sufficient to overcome the trade-off between generative quality and generative coherence, and most importantly results in instabilities with respect to hyperparameters controlling private latent space capacity, which calls for expensive model selection procedures in practical settings.

### 3.3 THE MMVAE+ OBJECTIVE

To enhance generative quality without compromising semantic coherence, we introduce the MM-VAE+ model, a novel variant of the MMVAE. Our model incorporates the idea of using factorized shared and modality-specific latent representations, and therefore makes the assumptions on the generative process and the variational approximation function presented in section 3.2. In order to ensure learning of a shared subspace while not incurring in a shortcut, auxiliary distributions for private features are used to facilitate the estimation of cross-modal reconstructions. The optimized objective is

$$\mathcal{L}_{\text{MMVAE+}}(\boldsymbol{x}_{1:M}) = \frac{1}{M} \sum_{m=1}^{M} \mathbb{E}_{\substack{q_{\phi_{\boldsymbol{z}_m}}(\boldsymbol{z}|\boldsymbol{x}_m) \\ q_{\phi_{\boldsymbol{w}_m}}(\boldsymbol{w}_m|\boldsymbol{x}_m) \\ \{\tilde{\boldsymbol{w}}_n \sim r_n(\boldsymbol{w}_n)\}_{n \neq m}}} \log \left( \frac{p_{\theta_m}(\boldsymbol{x}_m|\boldsymbol{z}, \boldsymbol{w}_m) p(\boldsymbol{z}) p(\boldsymbol{w}_m)}{q_{\Phi_{\boldsymbol{z}}}(\boldsymbol{z}|\boldsymbol{X}) q_{\phi_{\boldsymbol{w}_m}}(\boldsymbol{w}_m|\boldsymbol{x}_m)} \prod_{n \neq m} p_{\theta_n}(\boldsymbol{x}_n|\boldsymbol{z}, \tilde{\boldsymbol{w}}_n) \right)$$

(5)

where $r_1(\boldsymbol{w}_1), \ldots, r_M(\boldsymbol{w}_M)$ are auxiliary prior distributions on modality-specific features for each modality. [1]

Comparing with the objective in 4, the inferred modality-specific encoding is used *only* for self-reconstruction, and *not* for cross-modal reconstruction, for which the modality-specific code is sampled from the auxiliary prior. This crucial feature avoids the creation of a shortcut in the presence of modality-specific latent subspaces. In fact, cross-modal reconstructions are computed conditioning on a non-informative value for the modality-specific encoding, which forces the decoder to rely on the shared encoding $\boldsymbol{z}$ to reconstruct unobserved modalities. As a consquence, $\boldsymbol{z}$ encodes information useful to maximize reconstruction across modalities, i.e. shared information. Still, modality-specific subspaces can serve the purpose of modelling private variation, as private encoders are used for computing modality-specific codes for self-reconstruction.

The following lemma, for which we provide a proof in Appendix A, proves that MMVAE+ optimizes a valid ELBO and therefore belongs to the family of multimodal VAEs.

**Lemma 1.** *The MMVAE+ objective (Equation 5) is a valid lower bound on* $\log p_{\Theta}(\boldsymbol{x}_{1:M})$.

Note that while we obtain a valid ELBO for an arbitrary choice of auxiliary priors, we discuss our specific design choices in Appendix C. As in Shi et al. (2019), a multi-sample version of the objective can provide a tighter bound on the log-evidence. In Appendix D we discuss the details of the tighter version of the objective, along with the possibility to include a $\beta$ hyperparameter (Higgins et al., 2017) to weight latent space regularization.

## 4 EXPERIMENTS

We report experimental results for two challenging datasets introduced in previous work, namely PolyMNIST (Sutter et al., 2021) and Caltech Birds (CUB) Image-Captions (Shi et al., 2019; Wah et al., 2011). PolyMNIST is a synthetic dataset consisting of five modalities, depicting MNIST digits sharing the digit label, patched on random crops from five different background images, one for each modality. Each datapoint consists of five images, one per modality, where all the images share the digit label, but not the style of the handwriting, as MNIST samples are shuffled prior to the pairing. Shared information between modalities is therefore the digit label, while modality-specific information includes the style of the handwriting and the specific background image. The Caltech Birds (CUB) Image-Captions dataset consists of images of birds paired with matching linguistic descriptions. What makes this a challenging experiment is the large amount of modality-specific information for each modality, due to the different nature of the information sources. While previous work (Shi et al., 2019; 2021; Joy et al., 2022) has tackled a simplified version of this experiment by using pretrained ResNet-features, we train the models on actual images—a more realistic setting that was shown to be challenging for the predominant approaches (Daunhawer et al., 2022). All quantitative results are averaged over three independent seeds and we report standard deviations. Technical details on the experiments and additional results are available in Appendices F and G.

---

[1]Note that we refer to these as priors in the sense that they are not conditional on the data, but are effectively surrogate non-informative inference distributions for private features, which do not depend on the observed data samples, and are different from the prior distributions for private information in the assumed generative process $p(\boldsymbol{w}_1), \ldots, p(\boldsymbol{w}_M)$.

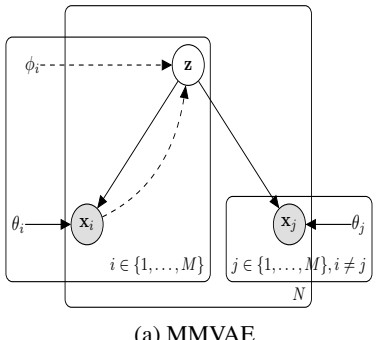

(a) MMVAE

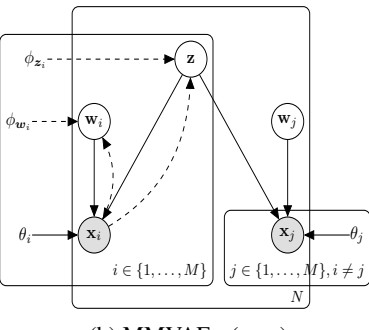

(b) MMVAE+ (ours)

Figure 2: Graphical models for comparison of MMVAE (left) and MMVAE+ (right). For both models, gray circles represent observed variables, white circles represent latent variables, and non-circled are the model parameters. Solid lines denote the generative process, while dashed arrows denote approximated posterior inference with encoders. Note that for MMVAE+ cross-modal reconstruction does not depend on inferred modality-specific information, thereby relying on the shared latent space $z$ to encode information useful across modalities, i.e. shared information. At the same time, maximizing self-reconstruction of each modality $m$ given the modality-specific $w_m$ code, encourages private variation to be encoded in modality-specific subspaces.

## 4.1 ENHANCED GENERATIVE QUALITY FOR SEMANTICALLY COHERENT GENERATION

This section shows our results on the PolyMNIST dataset comparing MMVAE+ to the predominant approaches, namely MMVAE (Shi et al., 2019), MVAE (Wu & Goodman, 2018) and MoPoE-VAE (Sutter et al., 2021). For completeness, we also compare with two recently proposed variations of MVAE and MMVAE, namely the MVTCAE (Hwang et al., 2021) and the mmJSD (Sutter et al., 2020), which both introduce additional regularization terms to the respective objectives. Qualitative results for conditional generation in Figure 3 illustrate the shortcomings of the existing approaches. The MVAE exhibits a lack of generative coherence, whereas the MMVAE and MoPoE-VAE both show a lack of sample diversity, resulting in poor generative quality. Even in this relatively simple setting, modality-specific information—such as the background details or the style of handwriting—tends to collapse to expected values for cross-modal generation. In contrast, the MMVAE+ shows a significantly better generative quality with an improved sample diversity and high semantic coherence. Comparing directly with the MMVAE, the results validate that our approach achieves enhanced generative quality without compromising semantic coherence.

For a quantitative comparison, we measure the performance in terms of generative coherence (Shi et al., 2019) and generative quality. The latter is estimated in terms of FID score (Heusel et al., 2017), a standard metric to evaluate sample quality for generative models in image domains. The scatterplots in Figure 4 show that only the MMVAE+ reaches high generative quality *and* high semantic coherence consistently over a large range of hyperparameter values, while other models underperform in one of the two performance criteria. Note that while MVTCAE and mmJSD both show some improvement over the predominant approaches they are based on, e.g. MVTCAE markedly improves generative coherence over the MVAE, they still exhibit poor performance in either conditional or unconditional genenration. Notably, the advantage of MMVAE+ persists for both unconditional and conditional generation. Details for unconditional generation for our approach, which assumes separate latent subspaces, are at the end of Appendix F.

## 4.2 COMPARING WITH ALTERNATIVE APPROACHES WITH SEPARATE LATENT SUBSPACES

In this section, we compare MMVAE+ with alternative variants of multimodal VAE objectives that adopt a latent space that factorizes in separate shared and modality-specific subspaces. In particular, we compare with the factorized MMVAE objective in 4, as well as with its alternative version proposed in Sutter et al. (2020), which extends the single-space mmJSD model to a factorized latent space. In addition we include DMVAE (Lee & Pavlovic, 2021) in the comparison, which implements a product-based objective with factorized latent representations. However, in contrast to the other compared models, its optimized objective is not provably an ELBO. Crucially, we find that all these

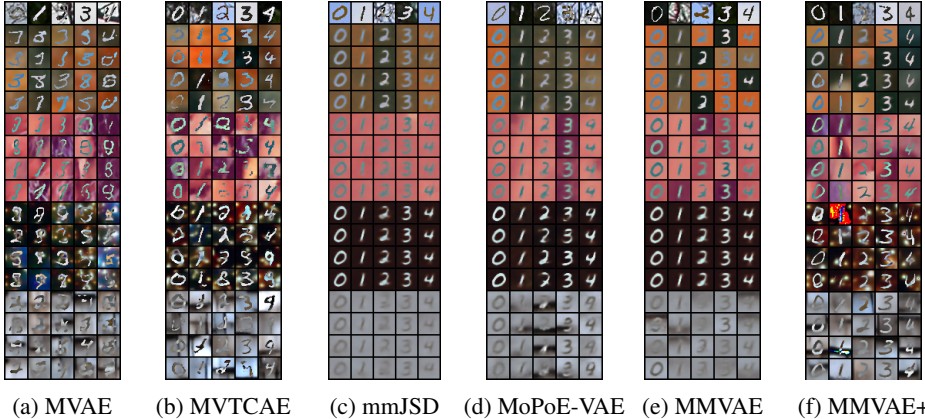

| (a) MVAE | (b) MVTCAE | (c) mmJSD | (d) MoPoE-VAE | (e) MMVAE | (f) MMVAE+ |

Figure 3: Qualitative results for conditional generation on the PolyMNIST dataset. The input sample from the first modality is shown in the top row, and below are four conditionally generated samples for each of the remaining modalities.

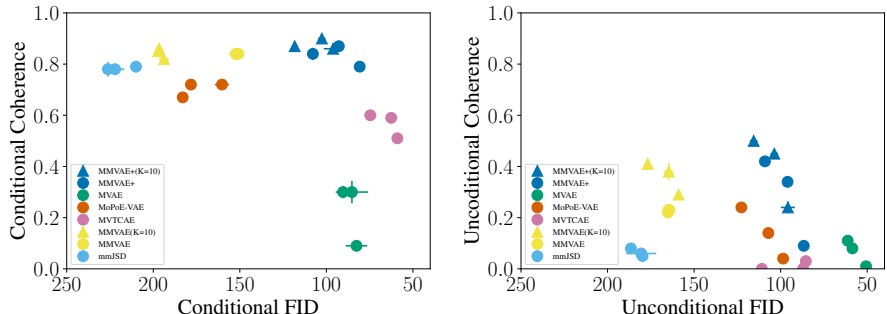

Figure 4: Comparison of generative quality (in terms of FID, lower is better) and generative coherence (higher is better). The left scatterplot evaluates conditional (cross-modal) generation and the right subplot quantifies unconditional generation performance. For each model, we show the results for a range of representative $\beta$ values (see Appendix F). An optimal model would lie in the top-right area of each scatterplot.

existing approaches exhibit a non-trivial problem: to improve generative coherence, they require strong constraining of modality-specific latent capacity, to avoid incurring in a shortcut where all information used by decoders is encoded independently in modality-specific subspaces. [2] In figure 5a we report an ablation for generative coherence for the compared models, varying the relative size of the modality-specific subspace for each modality with respect to the shared subspace. Adopting any of the existing approaches with separate latent subspaces, generative coherence quickly deteriorates as more capacity is allocated to private subspaces, reaching values close to random when private and shared subspaces have equal size. This behaviour can arise from the fact that the encoding from each modality-specific subspace is used to compute both self- and cross-modal reconstructions of the given modalities (see section 3.2). Therefore, with enough capacity, the modality-specific encoder can learn all information for the given modality (i.e., both shared and private features), which creates a shortcut where all information flows through the modality-specific subspaces and the shared subspace is ignored by the decoder. As additional evidence for this phenomenon, we report in Figure 5b latent classification accuracy (see Appendix E.2) for private subspaces, when varying private latent capacity. This metric shows the amount of shared information that a digit classifier can retrieve from private embeddings. As expected, as modality-specific dimensionality is augmented, an increasing amount of shared information is present in the corresponding encodings. It is evident from the showcased results that current approaches achieve sufficient coherence across generated

---

[2]Note that Wang et al. (2016) propose to use dropout during training, instead of reducing private latent dimensionality, to avoid incurring in a shortcut. We address this in Appendix I.

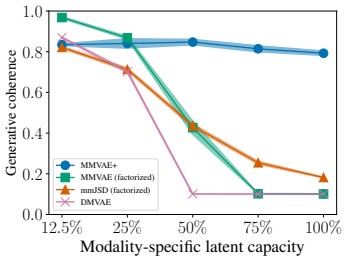 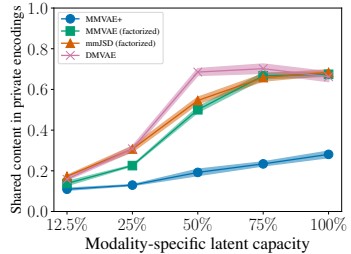 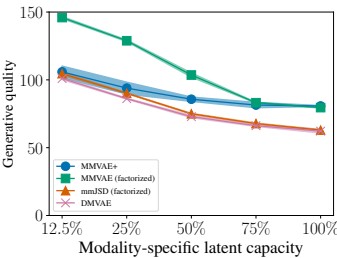

(a) Generative coherence (higher is better)

(b) Latent classification accuracy from modality-specific encodings (lower is better)

(c) Generative quality (in terms of FID score, lower is better)

Figure 5: Ablation for generative coherence, latent classification accuracy based on the private encodings, and generative quality, as a function of private latent dimensionality. The percentages on the x-axis indicate the relative size of any private subspace with respect to the shared subspace.

modalities, at the expense of carefully constraining the size of modality-specicic subspaces. This however has two undesirable effects. On one hand it inevitably limits modelling capacity for private information, which hampers generative quality, as showcased in figure 5c. On the other hand, and more importantly, achieving an adequate balance between the two performance criteria heavily depends on hyperparameter tuning, which calls for expensive model selection procedures in practical settings.

In contrast to the alternative approaches, the MMVAE+ proves to be significantly more robust with respect to hyperparameter values controlling the size of the modality-specific subspaces. In particular, our approach does not incur in a shortcut. Instead shows better performance, if jointly looking at generative quality and generative coherence, when shared and private latent subspaces for each modality have comparable capacities. Combining these results with the ones from the previous section, it is evident how our approach successfully exploits having additional modality-specific latent codes to enhance generative quality, without compromising semantic coherence.

## 4.3 CUB IMAGE-CAPTIONS

In this section, we test the MMVAE+ on the CUB Image-Captions (Shi et al., 2019; Netzer et al., 2011) dataset, and compare its performance with existing approaches in terms of conditional and unconditional generation, in order to validate our approach and test its impact on a complex real-world dataset. For the conditional caption-to-image generation (Figure 6), we again observe that modality-specific information tends to collapse to average values for mixture-based models, namely MMVAE, MoPoE-VAE and mmJSD, whereas MVAE and MVTCAE show convincing generative quality but a severe lack of coherence (cp. Daunhawer et al., 2022). Similar conclusions can be drawn for conditional image-to-caption generation (Figure 7). In contrast, MMVAE+ shows significantly better results for both caption-to-image and image-to-caption generation. Specifically, our model achieves high sample diversity for both the image and text modality. Hence, we demonstrate the effectiveness of our approach for more complex real-world data, where there is a sizeable amount of modality-specific information and significant heterogeneity across modalities.

|  | Conditional coherence | Conditional FID |
|---|---|---|
| MVAE | $0.271$ $(\pm 0.007)$ | $172.21$ $(\pm 39.61)$ |
| MVTCAE | $0.221$ $(\pm 0.007)$ | $208.43$ $(\pm 1.10)$ |
| mmJSD | $0.556$ $(\pm 0.158)$ | $262.80$ $(\pm 6.93)$ |
| MMVAE | $0.713$ $(\pm 0.057)$ | $232.20$ $(\pm 2.14)$ |
| MoPoE-VAE | $0.579$ $(\pm 0.158)$ | $265.55$ $(\pm 4.01)$ |
| MMVAE+ (ours) | $0.721$ $(\pm 0.090)$ | $164.94$ $(\pm 1.50)$ |

Table 1: Comparison of generative quality (in terms of FID, lower is better) and generative coherence (higher is better) for conditional caption-to-image generation on the CUB Image-Captions dataset.

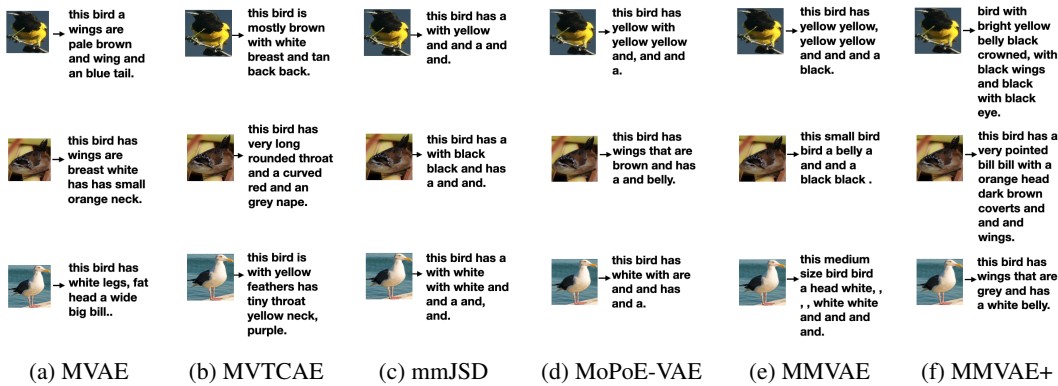

|  |  |  |  |  |  |
|---|---|---|---|---|---|
| (a) MVAE | (b) MVTCAE | (c) mmJSD | (d) MoPoE-VAE | (e) MMVAE | (f) MMVAE+ |

Figure 6: Conditional image-to-caption generation on the CUB Image-Caption dataset.

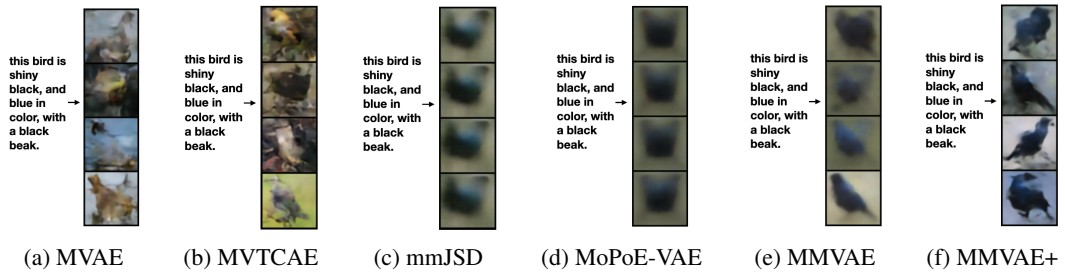

|  |  |  |  |  |  |
|---|---|---|---|---|---|
| (a) MVAE | (b) MVTCAE | (c) mmJSD | (d) MoPoE-VAE | (e) MMVAE | (f) MMVAE+ |

Figure 7: Conditional caption-to-image generation on the CUB Image-Caption dataset. For each model, the input caption is used to conditionally generate four images.

To quantitatively evaluate coherence, we design a new coherence proxy for the CUB Image-Captions dataset. While for the PolyMNIST dataset, the information shared between modalities always consists of the digit label, for the CUB Image-Captions dataset we do not have annotation for the shared content between modalities. Further, the ResNet-feature evaluation used in previous works (Shi et al., 2019; 2021) is not suitable to evaluate models that were trained on real images. Therefore, we design a coherence proxy for conditional caption-to-image generation based on the color of the described bird (see E.2 for details). Based on this metric, Table 1 provides a quantitative comparison of the models for caption-to-image generation in terms of generative coherence and generative quality. We again quantify visual quality of the produced images using the FID score. The quantitative results suggest that the MMVAE+ can model complex real-world datasets, where the predominant models fail to achieve satisfying results in at least one of the two performance criteria.

## 5 CONCLUSION

This work introduced MMVAE+, a new model in the family of multimodal VAEs, which derives from the MMVAE and considerably improves its generative quality without compromising semantic coherence. The proposed model uses separate shared and modality-specific subspaces and leverages auxiliary priors to facilitate the estimation of cross-modal reconstructions. We show that the model optimizes a valid multimodal ELBO and that it effectively encodes shared and modality-specific information in the respective subspaces. Our approach can be generalized and extended to other types of decompositions of the joint encoder, such as the mixture-of-products of experts. We leave this as a possibility for future work. On both synthetic and real-world data, we demonstrate the effectiveness of our approach for enhancing generative quality without compromising semantic coherence. Compared to previous approaches with separate latent subspaces, MMVAE+ is significantly more robust to hyperparameter values controlling the capacity of modality-specific subspaces. As such, MMVAE+ takes an important step towards the applicability of multimodal VAEs to complex, real-world multimodal datasets.

ACKNOWLEDGEMENTS

Emanuele Palumbo was supported by the grant #2021-911 of the Strategic Focal Area "Personalized Health and Related Technologies (PHRT)" of the ETH Domain (Swiss Federal Institutes of Technology). ID was supported by the SNSF grant #200021-188466. Special thanks to Carl Allen, Thomas Sutter and Yarden As for helpful discussions and/or feedback on the manuscript.

ETHICS STATEMENT

Given the nature of the experiments, and the datasets used we see no ethical concerns regarding this work.

REPRODUCIBILITY STATEMENT

We provide details about datasets, metrics, hyperparameters and architectures in the Appendix, which can aid reproducibility of our method along with the provided code.

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

## A    PROOFS

**Lemma 1.** *The MMVAE+ objective (Equation 5) is a valid lower bound on* $\log p_{\Theta}(\boldsymbol{X})$.

*Proof.* We assume the data generative process

$$p_{\Theta}(\boldsymbol{X}, \boldsymbol{z}, \boldsymbol{W}) = p(\boldsymbol{z}) \prod_{m=1}^{M} p_{\theta_m}(\boldsymbol{x}_m | \boldsymbol{z}, \boldsymbol{w}_m) p(\boldsymbol{w}_m),$$

where $\boldsymbol{z}$ encodes the shared content between modalities, and each $\boldsymbol{w}_m$ encodes modality-specific information for modality $m$. Note that shared latent information in $\boldsymbol{z}$ and modality-specific information for each modality $\boldsymbol{w}_1, \ldots, \boldsymbol{w}_M =: \boldsymbol{W}$ are assumed to be independent. We start by approximating posterior inference for the shared latent space with a mixture-of-experts variational encoder $q_{\Phi_{\boldsymbol{z}}}(\boldsymbol{z} | \boldsymbol{X}) = \frac{1}{M} \sum_{m=1}^{M} q_{\phi_{\boldsymbol{z}_m}}(\boldsymbol{z} | \boldsymbol{x}_m)$. With our assumptions the objective

$$\mathcal{L}_{\text{MMVAE}}(\boldsymbol{x}_{1:M}) = \frac{1}{M} \sum_{m=1}^{M} \mathbb{E}_{q_{\phi_{\boldsymbol{z}_m}}(\boldsymbol{z} | \boldsymbol{x}_m)} \left[ \log \frac{p(\boldsymbol{z}) p_{\theta_m}(\boldsymbol{x}_m | \boldsymbol{z}) \prod_{n \neq m} p_{\theta_n}(\boldsymbol{x}_n | \boldsymbol{z})}{q_{\Phi_{\boldsymbol{z}}}(\boldsymbol{z} | \boldsymbol{X})} \right]. \tag{6}$$

is a valid lower bound on the log-evidence $\log p_{\Theta}(\boldsymbol{X})$.

Note that this is the MMVAE objective (Shi et al., 2019), where for each term in the sum the encoding $\boldsymbol{z} \sim q_{\phi_{\boldsymbol{z}_m}}(\boldsymbol{z} | \boldsymbol{x}_m)$ is sampled from a unimodal encoder, and used to compute the conditional likelihood of all $M$ modalities: in particular both self-reconstruction likelihood $\log p_{\theta_m}(\boldsymbol{x}_m | \boldsymbol{z})$ and cross-modal reconstruction likelihoods $\log p_{\theta_n}(\boldsymbol{x}_n | \boldsymbol{z})$ for $n \neq m$ are evaluated. We derive the MMVAE+ objective by adopting different estimators for the two kinds of likelihood terms. In detail, we estimate $\log p_{\theta_m}(\boldsymbol{x}_m | \boldsymbol{z})$ using a variational encoder $q_{\phi_{\boldsymbol{w}_m}}(\boldsymbol{w}_m | \boldsymbol{x}_m)$ for modality-specific information and deriving the lower bound

$$\log p_{\theta_m}(\boldsymbol{x}_m | \boldsymbol{z}) \geq \mathbb{E}_{q_{\phi_{\boldsymbol{w}_m}}(\boldsymbol{w}_m | \boldsymbol{x}_m)} \left[ \log \frac{p_{\theta_m}(\boldsymbol{x}_m | \boldsymbol{w}_m, \boldsymbol{z}) p(\boldsymbol{w}_m)}{q_{\phi_{\boldsymbol{w}_m}}(\boldsymbol{w}_m | \boldsymbol{x}_m)} \right] \tag{7}$$

In contrast for cross-modal reconstruction likelihoods we propose the estimator

$$\log p_{\theta_n}(\boldsymbol{x}_n | \boldsymbol{z}) = \log \mathbb{E}_{\tilde{\boldsymbol{w}}_n \sim r_n(\boldsymbol{w}_n)} p_{\theta_n}(\boldsymbol{x}_n | \boldsymbol{z}, \tilde{\boldsymbol{w}}_n) \geq \mathbb{E}_{\tilde{\boldsymbol{w}}_n \sim r_n(\boldsymbol{w}_n)} \log p_{\theta_n}(\boldsymbol{x}_n | \boldsymbol{z}, \tilde{\boldsymbol{w}}_n) \tag{8}$$

using an auxiliary prior distribution $r_n(\boldsymbol{w}_n)$ specific to each target modality. Note the first step holds by definition of conditional expectation, while the second step follows from Jensen's inequality.

Plugging the derived expressions in 7 and 8 in equation 6, we recover the MMVAE+ objective

$$\mathcal{L}_{\text{MMVAE+}}(\boldsymbol{x}_{1:M}) = \frac{1}{M} \sum_{m=1}^{M} \mathbb{E}_{\substack{q_{\phi_{\boldsymbol{z}_m}}(\boldsymbol{z} | \boldsymbol{x}_m) \\ q_{\phi_{\boldsymbol{w}_m}}(\boldsymbol{w}_m | \boldsymbol{x}_m) \\ \{\tilde{\boldsymbol{w}}_n \sim r_n(\boldsymbol{w}_n)\}_{n \neq m}}} \log \left( \frac{p_{\theta_m}(\boldsymbol{x}_m | \boldsymbol{z}, \boldsymbol{w}_m) p(\boldsymbol{z}) p(\boldsymbol{w}_m)}{q_{\Phi_{\boldsymbol{z}}}(\boldsymbol{z} | \boldsymbol{X}) q_{\phi_{\boldsymbol{w}_m}}(\boldsymbol{w}_m | \boldsymbol{x}_m)} \prod_{n \neq m} p_{\theta_n}(\boldsymbol{x}_n | \boldsymbol{z}, \tilde{\boldsymbol{w}}_n) \right) \tag{9}$$

From the fact that equation 6 is an ELBO, and the fact that the derived expressions in 7 and 8 are lower bounds, we conclude that the MMVAE+ objective is an ELBO. $\square$

## B    ADDITIONAL THEORETICAL RESULTS

Here we prove a lemma, which formalizes the fact that with the MMVAE+ objective cross-modal reconstructions affect only the shared subspace but not the modality-specific subspaces. Specifically, in the MMVAE+ objective (Equation 5), the encoding $\boldsymbol{z} \sim q_{\phi_{\boldsymbol{z}_m}}(\boldsymbol{z} | \boldsymbol{x}_m)$ is used for the reconstruction of modality $m$, as well as for cross-modal reconstruction of the remaining modalities. As a consequence, the gradients for parameters $\phi_{\boldsymbol{z}_1}, \ldots, \phi_{\boldsymbol{z}_M}$ depend on the reconstruction likelihood of *all* modalities. Consequently, encoders $\{q_{\phi_{\boldsymbol{z}_m}}(\boldsymbol{z} | \boldsymbol{x}_m)\}_{m=1,\ldots,M}$ are optimized for $\boldsymbol{z}$ to contain information useful to reconstruct all modalities, i.e., *shared* information. In contrast, the encoding $\boldsymbol{w}_m \sim q_{\phi_{\boldsymbol{w}_m}}(\boldsymbol{w}_m | \boldsymbol{x}_m)$ is used only to compute the reconstruction likelihood of modality $m$. Hence, gradients for encoder parameters $\phi_{\boldsymbol{w}_1}, \ldots, \phi_{\boldsymbol{w}_M}$ only depend on self-reconstruction terms.

As a result, for optimized cross-modal reconstruction, all shared information needs to be contained in and predicted from $z$. On the other hand, private information for modality $m$ can be modelled in $w_m \sim q_{\phi_{w_m}}(w_m|x_m)$ as a consequence of optimizing self-reconstruction. As confirmed by our experimental results, our proposed objective leads to shared and private information being encoded in separate shared and modality-specific latent subspaces respectively.

**Lemma 2.** *Computed gradients for the MMVAE+ objective with respect to private encoder parameters $\phi_{w_1}, \ldots, \phi_{w_M}$ do not depend on cross-modal reconstruction likelihood terms, while gradients with respect to shared encoder parameters $\phi_{z_1}, \ldots, \phi_{z_M}$ depend on the reconstruction likelihood of all modalities.*

*Proof.* Computing the gradients for the MMVAE+ objective (Equation 5) with respect the private encoder network parameters $\phi_{w_m}$ for a given modality $m \in \{1, \ldots, M\}$ gives

$$
\nabla_{\phi_{w_m}} \mathcal{L}_{\text{MMVAE+}}(x_{1:M}) = \nabla_{\phi_{w_m}} \left( \mathbb{E}_{\substack{q_{\phi_{z_m}}(z|x_m) \\ q_{\phi_{w_m}}(w_m|x_m)}} \left[ \log \frac{p_{\theta_m}(x_m|z, w_m)p(w_m)}{q_{\phi_{w_m}}(w_m|x_m)} \right] \right)
$$

$$
= \nabla_{\phi_{w_m}} F_{\theta_m, \phi_{z_m}, \phi_{w_m}}(x_m)
$$

Note that the resulting function does not depend on decoder parameters $\{\theta_n\}_{n \neq m}$, i.e. does not depend on cross-modal reconstruction likelihoods. Computing the gradients with respect the shared encoder network parameters $\phi_{z_m}$ for a given modality $m \in \{1, \ldots, M\}$ instead gives

$$
\nabla_{\phi_{z_m}} \mathcal{L}_{\text{MMVAE+}}(x_{1:M}) = \nabla_{\phi_{z_m}} \left( \mathbb{E}_{\substack{q_{\phi_{z_m}}(z|x_m) \\ q_{\phi_{w_m}}(w_m|x_m) \\ \{\tilde{w}_n \sim r_n(w_n)\}_{n \neq m}}} \left[ \log \left( p_{\theta_m}(x_m|z, w_m)p(z) \prod_{n \neq m} p_{\theta_n}(x_n|z, \tilde{w}_n) \right) \right] \right.
$$

$$
\left. + \frac{1}{M} \sum_{r=1}^{M} \mathbb{E}_{z \sim q_{\phi_{z_r}}(z|x_r)} \left[ \log \frac{1}{q_{\Phi_z}(z|X)} \right] \right)
$$

$$
= \nabla_{\phi_{z_m}} G_{\Theta, \Phi_z, \phi_{w_m}}(X)
$$

which is a function of the reconstruction likelihood of all modalities given inference from modality $m$, through decoders parameterized by $\Theta = \{\theta_1, \ldots, \theta_M\}$. $\qquad\square$

## C  CHOICES OF AUXILIARY PRIORS

In this section we elaborate on the role of auxiliary prior distributions $r_1(w_1), \ldots, r_M(w_M)$ in our proposed objective, and the specific design choices for such distributions.

In our experiments, as auxiliary priors we choose zero-mean distributions with a variance parameter that is learnt at training time. Notably, the learnt variance is a separate parameter vector (for each private subspace) that does not affect the encoders. Moreover, the variance parameter, and hence the $r_1(w_1), \ldots, r_M(w_M)$ distributions are not computed conditioning on any specific data sample. Therefore having a variance parameter does not affect our results from lemma 1.

Having a learnt variance is a key component as it allows the auxiliary priors to adapt the effective value range of the private encoding for cross-modal reconstructions. Specifically, the prior can adapt the value range to serve as an indicator that informs the decoder about self-reconstructions versus cross-modal reconstructions. Empirically, we find support for this hypothesis, as each auxiliary prior tends to become a sharp distribution (i.e., approximately a delta function) in the course of training.

# D    INCLUDING A MULTI-SAMPLE ESTIMATOR AND A $\beta$ HYPERPARAMETER IN THE MMVAE AND MMVAE+ OBJECTIVES

## D.1    MULTI-SAMPLE MMVAE AND MMVAE+ OBJECTIVES

To tighten the bound on the log-evidence in 2, Shi et al. (2019) propose to use a multi-sample estimator. In fact the $K$-sample objective

$$\mathcal{L}_{\text{MMVAE}_\beta}^{ms}(\boldsymbol{x}_{1:M}) = \frac{1}{M}\sum_{m=1}^{M}\mathbb{E}_{\boldsymbol{z}^{1:K}\sim q_{\phi_m}(\boldsymbol{z}|\boldsymbol{x}_m)}\left[\log\frac{1}{K}\sum_{k=1}^{K}\frac{p_\Theta(\boldsymbol{X},\boldsymbol{z}^k)}{q_\Phi(\boldsymbol{z}^k|\boldsymbol{X})}\right]$$

is a tighter ELBO compared to (2) for $K > 1$, as it can be shown with Jensen's inequality. However, as multi-sample estimators can lead to undesirably high variance, Tucker et al. (2019) have proposed a doubly-reparameterized gradient estimator (DReG) to reduce the variance in estimated gradients for latent-variable models with multi-sample objectives. Shi et al. (2019) resort to this estimator in computing gradients for the multi-sample version of the MMVAE objective.

Analogously for the MMVAE+ objective, a tighter bound on the log-evidence can be enforced if one uses a multi-sample estimator. In fact the $K$-sample version of the objective in (5)

$$\mathcal{L}_{\text{MMVAE+}}^{ms}(\boldsymbol{x}_{1:M}) = \frac{1}{M}\sum_{m=1}^{M}\mathbb{E}_{\substack{\boldsymbol{z}_m^{1:K}\sim q_{\phi_{\boldsymbol{z}_m}}(\boldsymbol{z}|\boldsymbol{x}_m)\\ \boldsymbol{w}_m^{1:K}q_{\phi_{\boldsymbol{w}_m}}(\boldsymbol{w}_m|\boldsymbol{x}_m)\\ \{\tilde{\boldsymbol{w}}_n^{1:K}\sim r_n(\boldsymbol{w}_n)\}_{n\neq m}}}\log\frac{1}{K}\sum_{k=1}^{K}C_{\Phi,\Theta}(\boldsymbol{X},\boldsymbol{z}^k,\tilde{\boldsymbol{w}}_1^k,\dots,\boldsymbol{w}_m^k,\dots,\tilde{\boldsymbol{w}}_M^k)$$

(10)

with

$$C_{\Phi,\Theta}(\boldsymbol{X},\boldsymbol{z}^k,\tilde{\boldsymbol{w}}_1^k,\dots,\boldsymbol{w}_m^k,\dots,\tilde{\boldsymbol{w}}_M^k) = \frac{p_{\theta_m}(\boldsymbol{x}_m|\boldsymbol{z}^k,\boldsymbol{w}_m^k)p(\boldsymbol{z}^k)p(\boldsymbol{w}_m^k)}{q_{\Phi_{\boldsymbol{z}}}(\boldsymbol{z}^k|\boldsymbol{X})q_{\phi_{\boldsymbol{w}_m}}(\boldsymbol{w}_m^k|\boldsymbol{x}_m)}\prod_{n\neq m}p_{\theta_n}(\boldsymbol{x}_n|\boldsymbol{z}^k,\tilde{\boldsymbol{w}}_n^k)$$

is a tighter ELBO compared to (5) for $K > 1$, as it can be shown with Jensen's inequality.

## D.2    BALANCING RECONSTRUCTION AND REGULARIZATION WITH A $\beta$ HYPERPARAMETER

As common practice in the VAE literature, the objective in (1) can be rewritten as a sum of a reconstruction term and a KL-divergence term

$$\mathcal{L}_{\text{VAE}_\beta}(\boldsymbol{x}_{1:M}) = \mathbb{E}_{q_\Phi(\boldsymbol{z}|\boldsymbol{X})}\left[\log p_\Theta(\boldsymbol{X}|\boldsymbol{z})\right] - \beta D_{KL}(q_\Phi(\boldsymbol{z}|\boldsymbol{X})\parallel p(\boldsymbol{z}))$$

(11)

where a $\beta$ hyperparameter weighting the latter term is introduced to control smoothness of the latent space (Higgins et al., 2017). With the assumption of a mixture-of-experts joint encoder, one can in the same way rewrite the MMVAE objective as the sum of a reconstruction and a KL-divergence term, weighted with a $\beta$ hyperparameter

$$\mathcal{L}_{\text{MMVAE}_\beta}(\boldsymbol{x}_{1:M}) = \frac{1}{M}\sum_{m=1}^{M}\mathbb{E}_{q_{\phi_m}(\boldsymbol{z}|\boldsymbol{X})}\left[\log p_\Theta(\boldsymbol{X}|\boldsymbol{z})\right] - \beta D_{KL}(q_\Phi(\boldsymbol{z}|\boldsymbol{X})\parallel p(\boldsymbol{z}))$$

$$= \frac{1}{M}\sum_{m=1}^{M}\mathbb{E}_{q_{\phi_m}(\boldsymbol{z}|\boldsymbol{x}_m)}\left[\log\frac{p_\Theta(\boldsymbol{X}|\boldsymbol{z})(p(\boldsymbol{z}))^\beta}{(q_\Phi(\boldsymbol{z}|\boldsymbol{X}))^\beta}\right]$$

(12)

From the last expression, one can derive a multi-sample version of the MMVAE objective

$$\mathcal{L}_{\text{MMVAE}_\beta}^{ms}(\boldsymbol{x}_{1:M}) = \frac{1}{M}\sum_{m=1}^{M}\mathbb{E}_{\boldsymbol{z}^{1:K}\sim q_{\phi_m}(\boldsymbol{z}|\boldsymbol{x}_m)}\left[\log\frac{1}{K}\sum_{k=1}^{K}\frac{p_\Theta(\boldsymbol{X}|\boldsymbol{z}^k)(p(\boldsymbol{z}^k))^\beta}{(q_\Phi(\boldsymbol{z}^k|\boldsymbol{X}))^\beta}\right]$$

(13)

where the $\beta$ hyperparameter is introduced. Note that Jensen's inequality proves (13) is a tighter bound on the log-evidence compared to (12) for $K > 1$ and any fixed value of $\beta \geq 1$.

Analogously, we can derive a $\beta$-weighted version of the $K$-sample MMVAE+ objective

$$\mathcal{L}_{\text{MMVAE+}_\beta}(\boldsymbol{x}_{1:M}) = \frac{1}{M} \sum_{m=1}^{M} \mathbb{E}_{\substack{q_{\phi_{\boldsymbol{z}_m}}(\boldsymbol{z}|\boldsymbol{x}_m) \\ q_{\phi_{\boldsymbol{w}_m}}(\boldsymbol{w}_m|\boldsymbol{x}_m) \\ \{\tilde{\boldsymbol{w}}_n \sim r_n(\boldsymbol{w}_n)\}_{n \neq m}}} \log \left( \frac{p_{\theta_m}(\boldsymbol{x}_m|\boldsymbol{z}, \boldsymbol{w}_m)(p(\boldsymbol{z})p(\boldsymbol{w}_m))^\beta}{(q_{\Phi_{\boldsymbol{z}}}(\boldsymbol{z}|\boldsymbol{X})q_{\phi_{\boldsymbol{w}_m}}(\boldsymbol{w}_m|\boldsymbol{x}_m))^\beta} \prod_{n \neq m} p_{\theta_n}(\boldsymbol{x}_n|\boldsymbol{z}, \tilde{\boldsymbol{w}}_n) \right)$$

and the corresponding $K$-sample objective

$$\mathcal{L}_{\text{MMVAE+}_\beta}^{ms}(\boldsymbol{x}_{1:M}) = \frac{1}{M} \sum_{m=1}^{M} \mathbb{E}_{\substack{\boldsymbol{z}_m^{1:K} \sim q_{\phi_{\boldsymbol{z}_m}}(\boldsymbol{z}|\boldsymbol{x}_m) \\ \boldsymbol{w}_m^{1:K} q_{\phi_{\boldsymbol{w}_m}}(\boldsymbol{w}_m|\boldsymbol{x}_m) \\ \{\tilde{\boldsymbol{w}}_n^{1:K} \sim r_n(\boldsymbol{w}_n)\}_{n \neq m}}} \log \frac{1}{K} \sum_{k=1}^{K} D_{\Phi,\Theta}^\beta(\boldsymbol{X}, \boldsymbol{z}^k, \tilde{\boldsymbol{w}}_1^k, \ldots, \boldsymbol{w}_m^k, \ldots, \tilde{\boldsymbol{w}}_M^k)$$

with

$$D_{\Phi,\Theta}^\beta(\boldsymbol{X}, \boldsymbol{z}^k, \tilde{\boldsymbol{w}}_1^k, \ldots, \boldsymbol{w}_m^k, \ldots, \tilde{\boldsymbol{w}}_M^k) = \frac{p_{\theta_m}(\boldsymbol{x}_m|\boldsymbol{z}^k, \boldsymbol{w}_m^k)(p(\boldsymbol{z}^k)p(\boldsymbol{w}_m^k))^\beta}{(q_{\Phi_{\boldsymbol{z}}}(\boldsymbol{z}^k|\boldsymbol{X})q_{\phi_{\boldsymbol{w}_m}}(\boldsymbol{w}_m^k|\boldsymbol{x}_m))^\beta} \prod_{n \neq m} p_{\theta_n}(\boldsymbol{x}_n|\boldsymbol{z}^k, \tilde{\boldsymbol{w}}_n^k)$$

Note that in our experiments, when training the MMVAE and the MMVAE+ with the respective $K$-sample $\beta$-weighted objectives we make use of the DReG estimator (Tucker et al., 2019).

## E    TECHNICAL DETAILS FOR DATASETS AND METRICS

### E.1    DATASETS

In this work, we report experimental results on two multimodal datasets introduced in previous works, namely PolyMNIST (Sutter et al., 2021) and CUB Image-Captions (Shi et al., 2019; Netzer et al., 2011).

PolyMNIST is a synthetic dataset consisting of five image modalities. Each datapoint consists of five images depicting MNIST samples sharing the digit label, patched on random crops from five different background images, one for each modality. Shared information between modalities is the digit label, while private information comprises the background and the style of the handwriting, since MNIST digits are shuffled prior to the pairing.

The Caltech Birds (CUB) Image-Captions dataset consists of images of birds paired with matching linguistic descriptions. Making this experiment ambitious is the high amount of modality-specific information for each modality, due to the different nature of the information sources. Moreover, while for PolyMNIST the amount of shared information for each sample is fixed, always consisting of the digit label, here the amount of shared information present can vary from datapoint to datapoint. In fact, descriptions can be more or less detailed, leading to more or less shared information between modalities. This dataset presents a lot of challenges typical of a realistic experiment, which makes it a relevant setting to benchmark existing approaches.

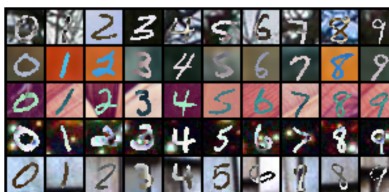 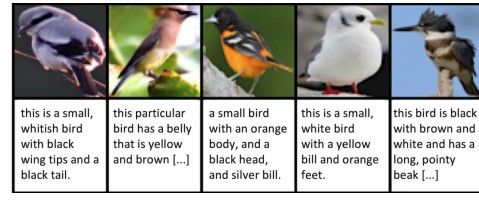

Figure 8: Illustrative samples from PolyMNIST (left) and CUB Image-Captions (right) respectively. Images taken from Shi et al. (2019) and Daunhawer et al. (2022).

### E.2    METRICS

**FID score**    In our experiments, to evaluate generative quality for image modalities we adopt the FID score (Heusel et al., 2017), a state-of-the-art metric to quantify visual sample quality for generative models in image domains, which has been shown to correlate well with human judgement. To compute FID scores we use the implementation from Seitzer (2020).

| white | yellow | blue | red | |
|---|---|---|---|---|
| $[0, 0, 120]$ | $[25, 50, 70]$ | $[90, 50, 70]$ | $[0, 50, 70]$ | $[159, 50, 70]$ |
| $[180, 18, 255]$ | $[35, 255, 255]$ | $[158, 255, 255]$ | $[15, 255, 255]$ | $[180, 255, 255]$ |
| **green** | **gray** | **brown** | **black** | |
| $[36, 50, 70]$ | $[0, 0, 50]$ | $[24, 255, 255]$ | $[0, 0, 0]$ | |
| $[89, 255, 255]$ | $[180, 18, 120]$ | $[16, 50, 70]$ | $[180, 255, 50]$ | |

Table 2: HSV (Hue, Saturation, Value) color ranges used to classify pixels in color classes. Each pixel in HSV color coding is represented by three values. Lower and upper limit(s) for each color class range(s) are reported below each color class label. Note that to include all tonalities of *red* one has to consider two distinct ranges.

**Generative coherence: PolyMNIST**    To compute generative coherence in the PolyMNIST experiment, as done in previous works (Shi et al., 2019; Sutter et al., 2021; Daunhawer et al., 2022), we use the training samples for each single modality to train a classifier for the digit label. The resulting classifiers are then used to obtain a quantitative measure for conditional and unconditional generative coherence. To measure conditional generative coherence from a given input modality $m$ to a given target modality $n$, we feed the conditionally generated sample to the classifier for the target modality $n$, then compare the predicted digit label with the true digit label. The matching rate for the whole test set measures conditional generative coherence from modality $m$ to modality $n$. To obtain a single representative metric for PolyMNIST, we compute the average coherence for each target modality, and then average the five obtained results. For unconditional generation, coherence is measured by feeding jointly-generated samples for each modality to the corresponding trained classifier. Then the measure for coherence is the rate with which all classifiers output the same predicted digit label on the total number of generations.

**Generative coherence: CUB Image-Captions** To have a quantitative assessment of semantic coherence in the CUB Image-Captions experiment we design a novel generative coherence metric for caption-to-image generation. In detail, we construct eight captions of the form *this bird is completely* [*color*], where [*color*] takes values in the set {*white, yellow, red, blue, green, grey, brown, black*}. We divide the HSV color coding range according to these eight color classes (see Table 2). Then, for each of the starting captions we generate ten images. For each generated image we count the pixels belonging to each color class. We label an image as *coherent* if the color for the starting caption is among the two color classes with the highest pixel count in the image. We take into account the *two* color classes with the highest count, as the highest count for some images might be the background color rather than the color of the depicted bird. Finally, the ratio of *coherent* images over the total number of generated images is our coherence metric.

We want to point out that, while Shi et al. (2019) propose a quantitative metric for the CUB Image-Captions dataset, this metric is effective only when dealing with the simpler version of this experiment that uses pretrained ResNet-features, rather than training on actual images. Shi et al. (2019) train the model on ResNet embeddings rather than real images: for image generation the model actually generates an embedding, that is matched with the nearest-neighbour embedding among all embeddings of the images in the original data. Therefore what the model outputs in an image belonging to the test set. This results in having high-quality images as output, but no generation in pixel space happens, which makes the task easier. The coherence metric used by Shi et al. (2019) relies on the fact that generated images are essentially taken from the dataset itself, and therefore come from the same distribution as the data. So, cross-generated images can be fed to the pre-trained ResNet used at training to compute a meaningful embedding to be compared with the embedding for the starting image. In our case setting instead, images are generated directly in pixel space, which means there would be a distribution shift between the generated images and the starting images (e.g. generated images are blurrier). With such a distribution shift, feeding generated images to the ResNet encoders trained on original data does not yield meaninful results, which invalidates such a metric for our setting.

**Latent classification accuracy**    In the PolyMNIST setting, where we have annotations for the shared information between modalities, i.e. the digit label, we use latent classification accuracy (Sutter et al., 2021; Daunhawer et al., 2022; Shi et al., 2019) as a proxy for the amount of shared

information encoded in a given latent subspace. To compute this metric, a digit-classifier is trained on the latent embeddings and its accuracy on the test set is a proxy for how much information about the shared digit content is encoded in the latent code. For our results in section 4.2, where we aim to have a measure of how much shared information is present in modality-specific subspaces, we train a digit classifier on each of the five modality-specific latent embeddings, and then average the test accuracies of the five classifiers to obtain a single metric.

## F  TECHNICAL DETAILS FOR THE EXPERIMENTS

**PolyMNIST (results in Section** 4.1 **and Appendix** $G$**)**     Following the suggestions by the authors in Shi et al. (2019) to reach best performance on image-to-image datasets, we train the MMVAE assuming Laplace priors, likelihoods and posteriors, constraining their scaling across the D dimensions to sum to D. For compatibility, we use the same settings for the MMVAE+. In our experiments we set the size of the shared latent subspace for MMVAE+ to 32 dimensions. In order not to induct any bias, and have equal shared and private latent capacity for each modality, we also set the size of each modality-specific latent subspace to 32 dimensions. For a fair comparison with the MMVAE, we set the size of the latent space to be equal to the total latent capacity of MMVAE+, namely 160 dimensions. When training both models with a multi-sample estimator in the objective, we reduce the size of the latent space of MMVAE to 64 dimensions, due to main memory constraints. MM-VAE and MMVAE+ are trained for 50 epochs when trained the $K$-sample version of the respective objectives, using $K = 10$ samples, while they are trained for 150 epochs when trained without multi-sample estimators. We train the mmJSD model again using a latent space of 160 dimensions, and for 150 epochs.

For MVAE, MoPoE-VAE, and MVTCAE, we use the settings in the work of Daunhawer et al. (2022) and Hwang et al. (2021), in order to obtain best performance in this experimental setting. We assume Laplace likelihoods and resort to Gaussian priors and posteriors which enable to compute product distributions in closed form. We train both MVAE and MoPoE-VAE for 500 epochs with a latent space of size 512, while MVTCAE (Hwang et al., 2021) is trained for 300 epochs, with the same latent space size.

We compare all models for a range of representative values of the regularization hyperparameter $\beta \in \{1.0, 2.5, 5.0\}$. Due to numerical instabilities the MVAE could not be trained with $\beta = 5.0$, and therefore for this model we set $\beta = 3.0$ as the last value of the regularization hyperparameter. Finally, for all compared models we use the same ResNet encoder and decoder networks, for each modality.

**PolyMNIST (results in Section** 4.2**)** For a fair comparison between the models, we train the MM-VAE using the objective in 4, the mmJSD with the objective assuming factorized latent space (Sutter et al., 2020), and the MMVAE+, using the same ResNet encoder and decoder networks for each modality, assuming Laplace likelihoods, and training all models for 150 epochs, without resorting to multi-sample estimators for any of the models. Note that the size of the shared latent subspace is always fixed to 32 dimensions.

**CUB Image-Captions**    On this experiment, for all compared models, we assume Laplace and one-hot categorical likelihood distributions for images and captions respectively, with Gaussian priors and posteriors, and set the latent space capacity to 64 dimensions. The size of private latent subspaces for MMVAE+ is chosen so that, for both modalities, summing the size of the private subspace and the shared subspace also amounts to 64 dimensions. We train MVAE, MoPoE-VAE, MVTCAE and mmJSD on this experiment for 150 epochs, with the settings to replicate best performance used in Daunhawer et al. (2022). In particular, ResNet encoder and decoder networks are used for both image and text modalities. MMVAE and MMVAE+ are trained for 50 epochs with the $K$-sample version of the respective objectives and $K = 10$, using the DReG estimator. Both models use Resnet network architectures for the image modality and CNNs for the text modality.

**General specifics for the experiments**    All models are trained using the Adam optimizer (Kingma & Ba, 2014). We choose a learning rate of 5e-4 for MVAE and MoPoE-VAE, following Daunhawer et al. (2022), while for MMVAE and MMVAE+ we choose 1e-3. mmJSD, and MVTCAE are also trained with learning rate 1e-3, as suggested in the respective implementations by the authors. All quantitative results are averaged over three seeds, and reported with standard deviations. Finally, we

want to specify that when comparing unconditional coherence in both experiments, for our approach which has separate latent subspaces, we sample the shared latent $z$ from its prior $p(z)$, and the modality-specific latent $w_m$ for each modality $m$ from the respective prior $p(w_m)$.

# G ADDITIONAL RESULTS

## G.1 POLYMNIST

**Detailed quantitative model comparison** In Table 3 we report the numerical values for each metric in the quantitative comparison in Section 4.1. These results show that only the MMVAE+ reaches high generative quality *and* high semantic coherence across different $\beta$ values, for both conditional and unconditional generation. Alternative models instead markedly underperform in one of the two criteria, or achieve convincing scores only for conditional *or* unconditional generation. The latter is the case with MVTCAE. In fact, while being an important improvement over the MVAE for conditional generation with greatly superior coherence, its performance markedly drops for unconditional generation, with extremely low values for coherence.

**Conditional generation** In figures 11 and 10 we report additional and more extensive qualitative results for the comparison between MMVAE+, MMVAE, MVAE, MVTCAE, mmJSD and MoPoE-VAE in section 4.1 for conditional generation.

**Latent classification accuracy** While conditional generation results showcased in Section 4.1 already offer sufficient evidence in this regard, here we show yet other results supporting the validity of our modelling assumptions where the shared latent subspace and the private latent subspace contain disentangled shared and modality-specific information respectively for each modality. In detail, we use latent classification accuracy (see Appendix E.2) to prove that the MMVAE+ achieves factorization between shared information in $z$ and private information in $w_1, \ldots, w_M$. For an ablation, similarly to section 4.2, we vary the proportion of latent dimensions allocated to the shared and the private subspace: with the shared subspace fixed to 32 dimensions, we vary the number of dimensions for each private subspace between $4, 8, 16, 24, 32$. Hence we go from having $12.5\%$ of the shared latent capacity allocated to the $w_m$ subspace for each modality $m$, to up to the two subspaces having the same size. The showcased results in Figure 12 indicate that the $z$ subspace successfully encodes shared content across modalities, consistently throughout the ablation, which validates our modelling assumptions for MMVAE+, showing they are robust to variation in hyperparameters. The accuracies obtained with classifiers trained on private embeddings are, as expected, markedly low. These results offer additional empirical evidence demonstrating that MMVAE+ learns representations that factorize into shared and private subspaces consistently with respect to changes in hyperparameters.

**Additional baseline results** As an additional baseline to compare our approach with, we include results obtained having the same assumptions for the generative model as in section 3.2, but assuming a product-of-experts joint encoder. The resulting objective essentially extends the MVAE (Wu & Goodman, 2018) to have separated shared and modality-specific latent subspaces. The optimized objective is

$$\mathcal{L}_{\text{MVAE}}^{f}(\boldsymbol{X}) = \mathbb{E}_{\substack{q_{\Phi_z}(\boldsymbol{z}|\boldsymbol{X}) \\ q_{\phi_{\boldsymbol{w}_1}}(\boldsymbol{w}_1|\boldsymbol{x}_1) \\ \vdots \\ q_{\phi_{\boldsymbol{w}_M}}(\boldsymbol{w}_M|\boldsymbol{x}_M)}} \left[ \log \frac{p_\Theta(\boldsymbol{X}, \boldsymbol{z}, \boldsymbol{W})}{q_{\Phi_z}(\boldsymbol{z}|\boldsymbol{X}) \prod_{m=1}^{M} q_{\phi_{\boldsymbol{w}_m}}(\boldsymbol{w}_m|\boldsymbol{x}_m)} \right] \tag{14}$$

where

$$q_{\Phi_z}(\boldsymbol{z}|\boldsymbol{X}) = \prod_{m=1}^{M} q_{\phi_{\boldsymbol{z}_m}}(\boldsymbol{z}|\boldsymbol{x}_m)$$

As it is expected and can be seen in Figure 9, this additional baseline model also proves to suffer from the shortcut problem, as the other existing multimodal VAEs with separate shared and modality-specific latent subspaces.

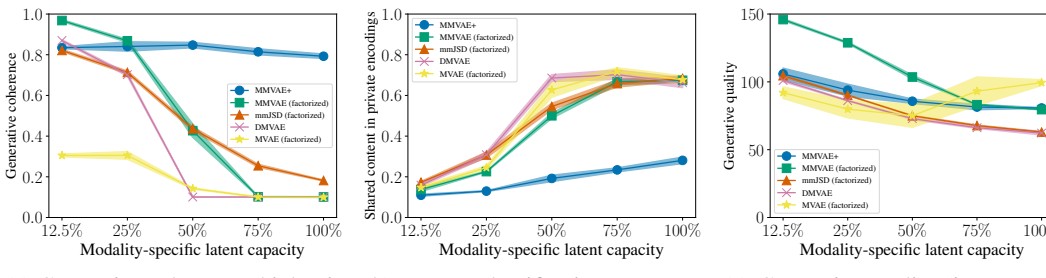

(a) Generative coherence (higher is better)

(b) Latent classification accuracy from modality-specific encodings (lower is better)

(c) Generative quality (in terms of FID score, lower is better)

Figure 9: Results for the ablation in Section 4.2 including the factorized version of MVAE (Wu & Goodman, 2018) as an additional baseline.

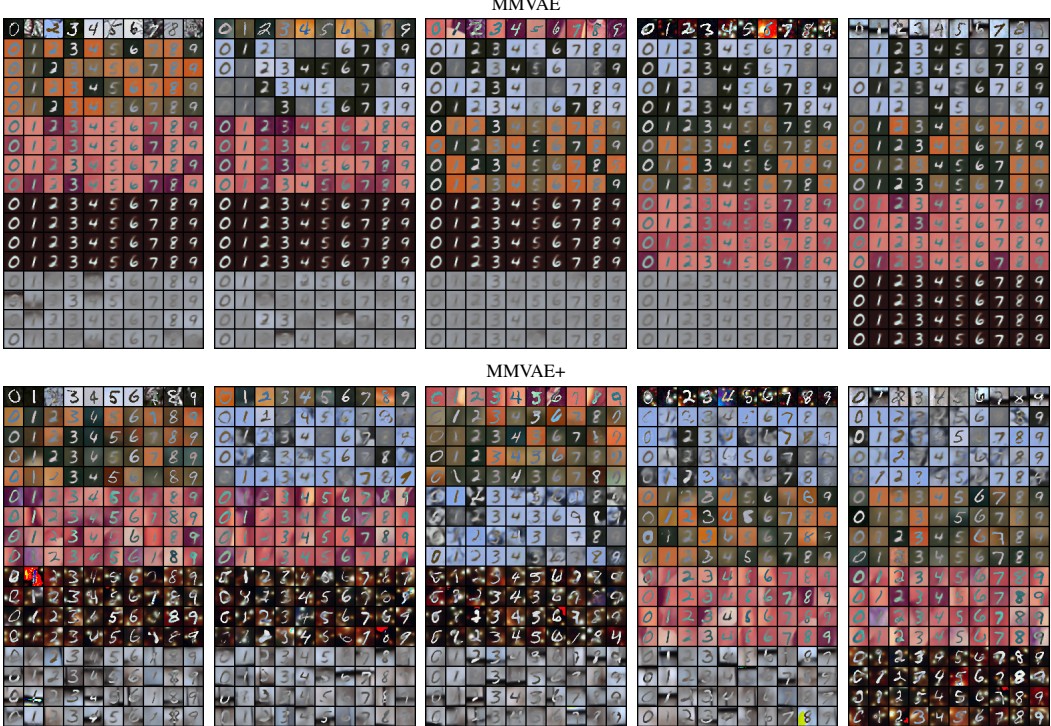

Figure 10: Conditional generation qualitative results for MMVAE and MMVAE+ on the PolyM-NIST dataset. For each different starting modality, we show input samples in the top row, and four instances of conditional generation for the remaining target modalities.

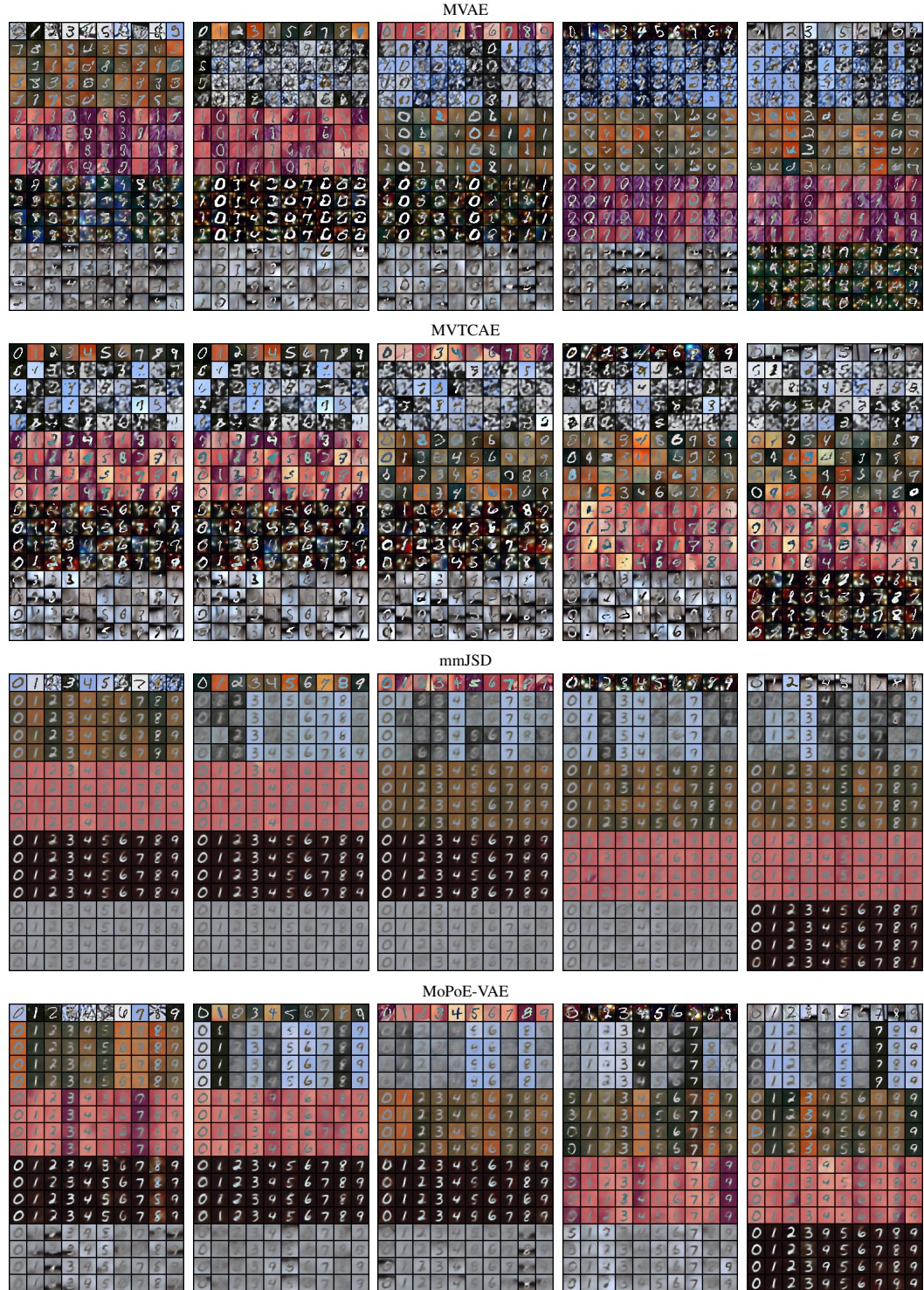

Figure 11: Conditional generation qualitative results for MVAE, MVTCAE, mmJSD and MoPoE-VAE on the PolyMNIST dataset. For each different starting modality, we show input samples in the top row, and four instances of conditional generation for the remaining target modalities.

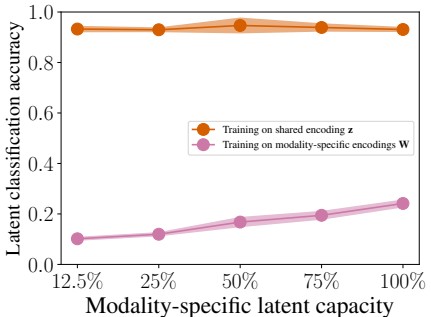

Figure 12: MMVAE+: latent classification accuracy computed on the shared latent embedding $z$ and the modality-specific latent embeddings $W = w_1, \ldots, w_M$, as a function of relative capacity of modality-specific subspaces. Note that this result validates our approach, which consistently encodes shared information across modalities in $z$, while this information is not present in $W$.

| $\beta = 1.0$ | Unconditional | | Conditional | |
|---|---|---|---|---|
| | FID | Coherence | FID | Coherence |
| MVAE | $50.65_{(\pm0.72)}$ | $0.007_{(\pm0.001)}$ | $82.59_{(\pm6.22)}$ | $0.093_{(\pm0.009)}$ |
| MVTCAE | $110.85_{(\pm2.61)}$ | $0.000_{(\pm0.000)}$ | $58.98_{(\pm0.62)}$ | $0.509_{(\pm0.006)}$ |
| mmJSD | $179.76_{(\pm2.97)}$ | $0.054_{(\pm0.011)}$ | $209.98_{(\pm1.26)}$ | $0.785_{(\pm0.023)}$ |
| MoPoE-VAE | $98.56_{(\pm1.32)}$ | $0.037_{(\pm0.002)}$ | $160.29_{(\pm4.12)}$ | $0.723_{(\pm0.006)}$ |
| MMVAE | $165.17_{(\pm3.40)}$ | $0.222_{(\pm0.019)}$ | $152.11_{(\pm4.11)}$ | $0.837_{(\pm0.004)}$ |
| MMVAE(K=10) | $158.95_{(\pm1.79)}$ | $0.292_{(\pm0.015)}$ | $196.62_{(\pm1.99)}$ | $0.865_{(\pm0.006)}$ |
| MMVAE+ | $86.64_{(\pm1.04)}$ | $0.095_{(\pm0.020)}$ | $80.75_{(\pm0.18)}$ | $0.796_{(\pm0.010)}$ |
| MMVAE+ (K=10) | $95.81_{(\pm4.04)}$ | $0.244_{(\pm0.023)}$ | $96.11_{(\pm5.36)}$ | $0.859_{(\pm0.012)}$ |

| $\beta = 2.5$ | Unconditional | | Conditional | |
|---|---|---|---|---|
| | FID | Coherence | FID | Coherence |
| MVAE | $58.53_{(\pm0.12)}$ | $0.080_{(\pm0.006)}$ | $85.23_{(\pm9.37)}$ | $0.298_{(\pm0.044)}$ |
| MVTCAE | $87.07_{(\pm0.89)}$ | $0.003_{(\pm0.000)}$ | $62.55_{(\pm1.30)}$ | $0.591_{(\pm0.004)}$ |
| mmJSD | $180.55_{(\pm8.67)}$ | $0.060_{(\pm0.010)}$ | $222.09_{(\pm5.34)}$ | $0.778_{(\pm0.003)}$ |
| MoPoE-VAE | $107.11_{(\pm0.780)}$ | $0.141_{(\pm0.005)}$ | $178.27_{(\pm2.01)}$ | $0.720_{(\pm0.008)}$ |
| MMVAE | $164.71_{(\pm3.17)}$ | $0.232_{(\pm0.010)}$ | $150.83_{(\pm2.69)}$ | $0.844_{(\pm0.010)}$ |
| MMVAE (K=10) | $164.64_{(\pm2.84)}$ | $0.379_{(\pm0.034)}$ | $197.24_{(\pm1.87)}$ | $0.853_{(\pm0.007)}$ |
| MMVAE+ | $96.01_{(\pm2.10)}$ | $0.344_{(\pm0.013)}$ | $92.81_{(\pm0.78)}$ | $0.869_{(\pm0.013)}$ |
| MMVAE+ (K=10) | $103.67_{(\pm2.22)}$ | $0.446_{(\pm0.009)}$ | $102.60_{(\pm2.17)}$ | $0.900_{(\pm0.002)}$ |

| $\beta = 5.0$ | Unconditional | | Conditional | |
|---|---|---|---|---|
| | FID | Coherence | FID | Coherence |
| MVAE | $61.25_{(\pm0.40)}$ | $0.112_{(\pm0.010)}$ | $90.37_{(\pm3.20)}$ | $0.301_{(\pm0.024)}$ |
| MVTCAE | $85.43_{(\pm2.80)}$ | $0.029_{(\pm0.001)}$ | $74.61_{(\pm3.41)}$ | $0.604_{(\pm0,004)}$ |
| mmJSD | $186.49_{(\pm2.89)}$ | $0.076_{(\pm0.018)}$ | $226.20_{(\pm2.91)}$ | $0.784_{(\pm0.029)}$ |
| MoPoE-VAE | $122.68_{(\pm1.96)}$ | $0.238_{(\pm0.001)}$ | $182.99_{(\pm1.96)}$ | $0.673_{(\pm0.002)}$ |
| MMVAE | $164.29_{(\pm2.97)}$ | $0.229_{(\pm0.017)}$ | $152.11_{(\pm3.18)}$ | $0.839_{(\pm0.010)}$ |
| MMVAE (K=10) | $176.79_{(\pm1.32)}$ | $0.407_{(\pm0.022)}$ | $193.79_{(\pm2.83)}$ | $0.817_{(\pm0.016)}$ |
| MMVAE+ | $109.08_{(\pm1.41)}$ | $0.421_{(\pm0.006)}$ | $107.78_{(\pm0.88)}$ | $0.8365_{(\pm0.023)}$ |
| MMVAE+ (K=10) | $115.57_{(\pm2.22)}$ | $0.501_{(\pm0.004)}$ | $118.30_{(\pm1.42)}$ | $0.868_{(\pm0.005)}$ |

Table 3: Comparison of model performance for generative quality (in terms of FID, lower is better) and generative coherence (higher is better). Both conditional (cross-modal) generation and unconditional generation performance are evaluated. Each table shows the results for the compared models for a different $\beta$ value. Note that values in the table relative to $\beta = 5.0$ for MVAE are actually computed with $\beta = 3.0$ due to numerical instabilities.

## G.2 CUB IMAGE-CAPTIONS

**Image quality for generation** In figures 13 and 15 we report unconditional and conditional image generation results for the compared models on the CUB Image-Captions dataset. Image generation results for MMVAE, mmJSD, and MoPoE-VAE in this complex experiment are characterized by the presence a considerable number of generated samples for which modality-specific features are collapsed to average values, resulting in blurred images. MMVAE+ results on the contrary do not exhibit signs of this pattern, which indicates good modelling of modality-specific information even in this complex setting, compared to the other mixture-based approaches. For the MVAE and MVT-CAE, product-based approaches, we witness instead the presence of significant variation in produced images, but low generative coherence comparing with our approach. In figure 16 we also show the generated samples resulting from our computation of the coherence score for this experiment, as described in appendix E.2.

**Image-to-caption generation** In figure 14 we showcase conditional image-to-caption results for the compared models, showing the MMVAE+ is the only approach achieving high quality for the produced sentences, which are also fairly coherent in attributes such as the colors present in the bird.

**Additional comparisons** Section 4.2 shows the central result for the comparison between MM-VAE+ and the recently proposed DMVAE (Lee & Pavlovic, 2021), which also employs factorized latent spaces. The fact that even in relatively simple datasets such as PolyMNIST, alternative models with factorized representations heavily suffer from the shortcut problem, already clearly separates our work from previous approaches. However, for completeness we also test this model in the CUB Image-Captions experiment, ensuring compatibility with MMVAE+ in architectures, hyperparameters, and other implementation details. We show qualitative and quantitative results for DMVAE on the CUB Image-Captions experiment in figure 17 and table 4 respectively. A qualitative and quantitative comparison of results clearly shows MMVAE+ achieves better results than DMVAE in this experiment.

|  | Conditional coherence | Conditional FID |
|---|---|---|
| DMVAE | $0.487$ ($\pm 0.022$) | $179.23$ ($\pm 6.99$) |
| MMVAE+ (ours) | $0.721$ ($\pm 0.090$) | $164.94$ ($\pm 1.50$) |

Table 4: Quantitative performance of DMVAE on the CUB Image-Captions experiment, compared with MMVAE+, in terms of conditional coherence (higher is better) and conditional FID (lower is better).

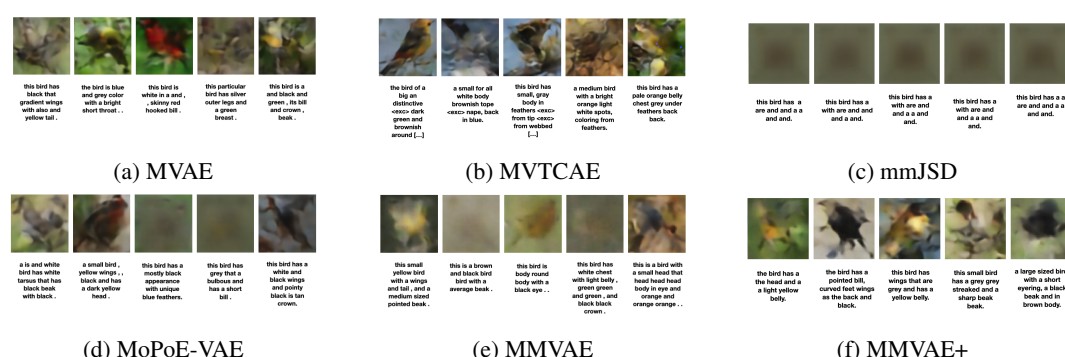

Figure 13: Qualitative results for MVAE, MVTCAE, mmJSD, MoPoE-VAE, MMVAE and MM-VAE+ for unconditional generation on the CUB Image-Captions dataset.

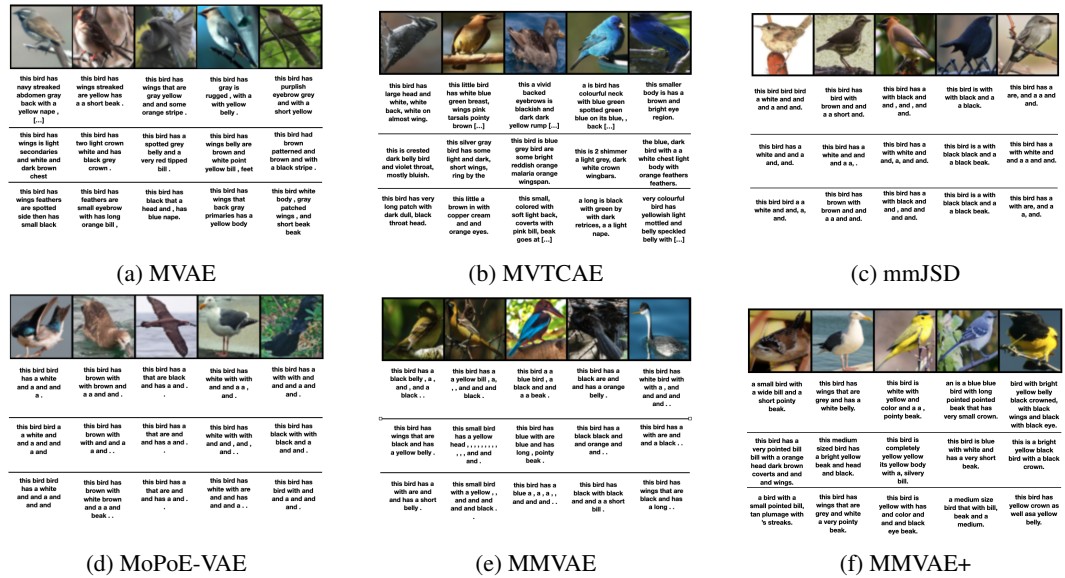

Figure 14: Qualitative results for MVAE, MVTCAE, mmJSD, MoPoE-VAE, MMVAE and MM-VAE+ for image-to-caption generation on the CUB Image-Captions dataset.

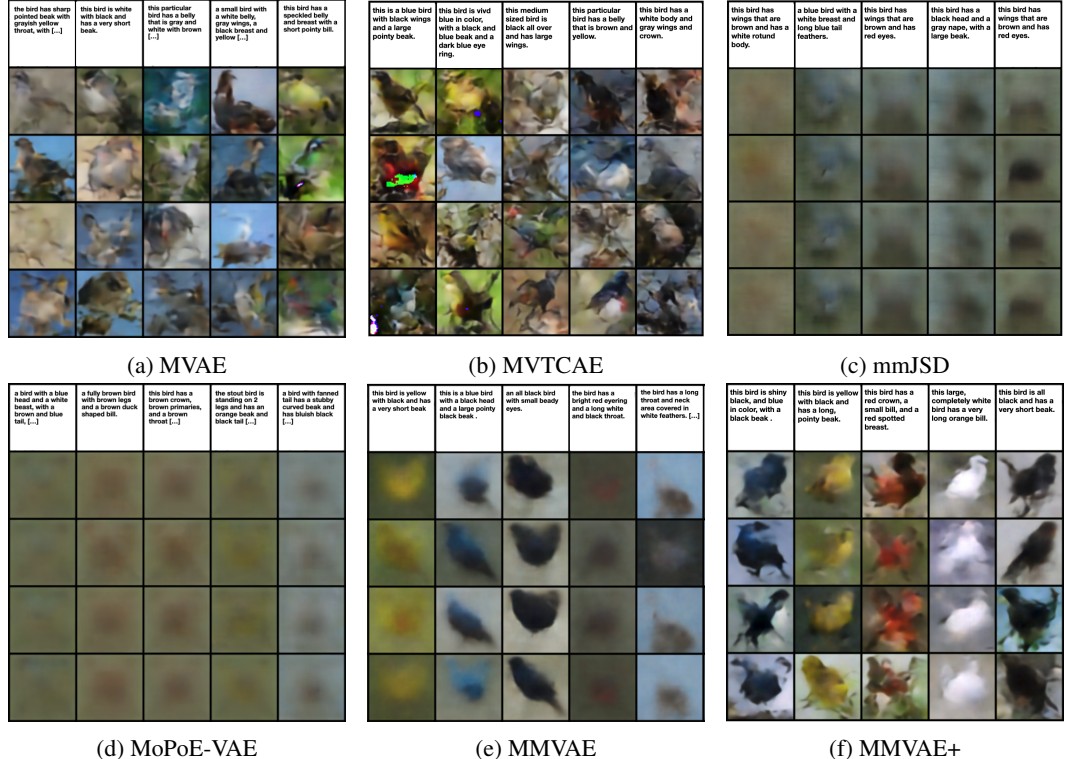

Figure 15: Qualitative results for MVAE, MVTCAE, mmJSD, MoPoE-VAE, MMVAE and MM-VAE+ for caption-to-image generation on the CUB Image-Captions dataset.

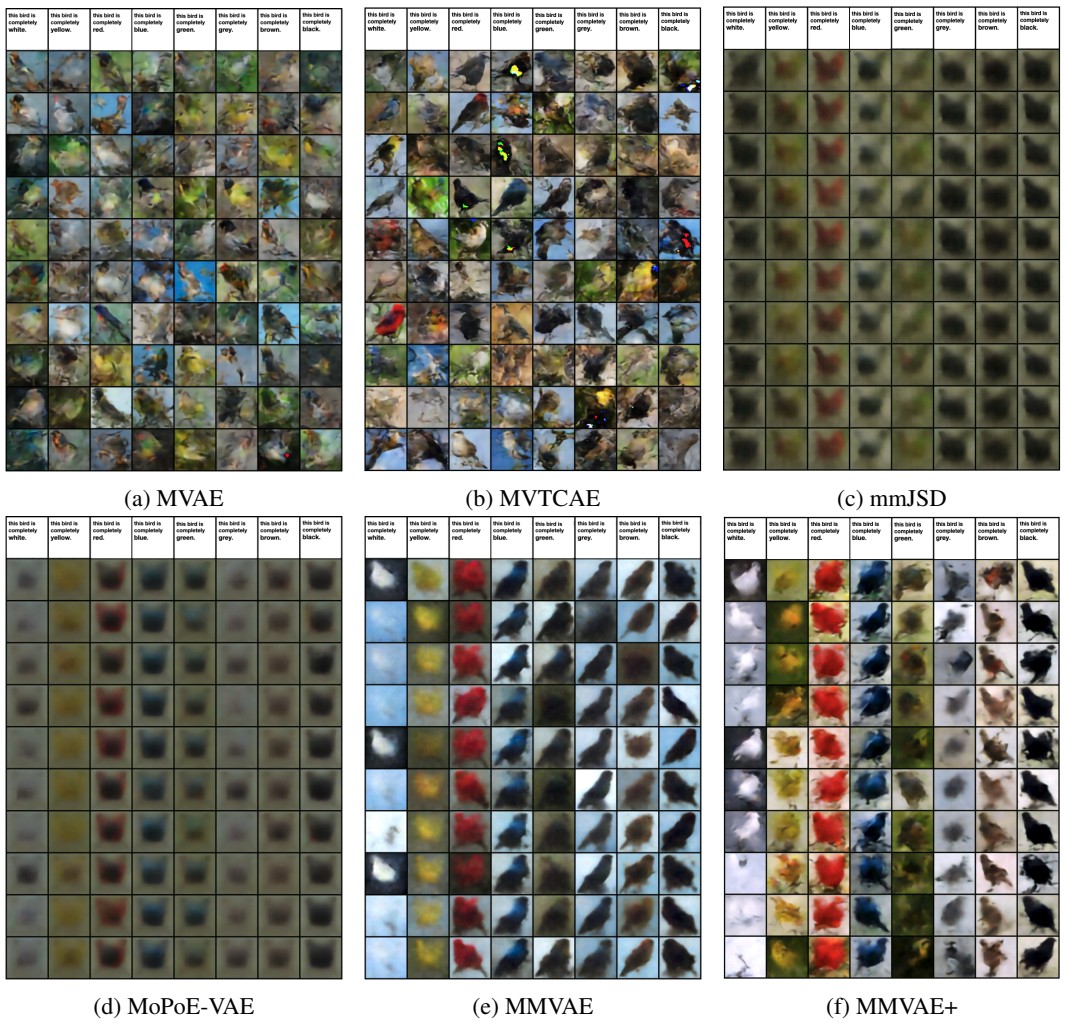

Figure 16: Qualitative results for the compared models for the caption-to-image generation procedure used for an estimation of generative coherence for the CUB Image-Captions experiment as described in appendix E.2

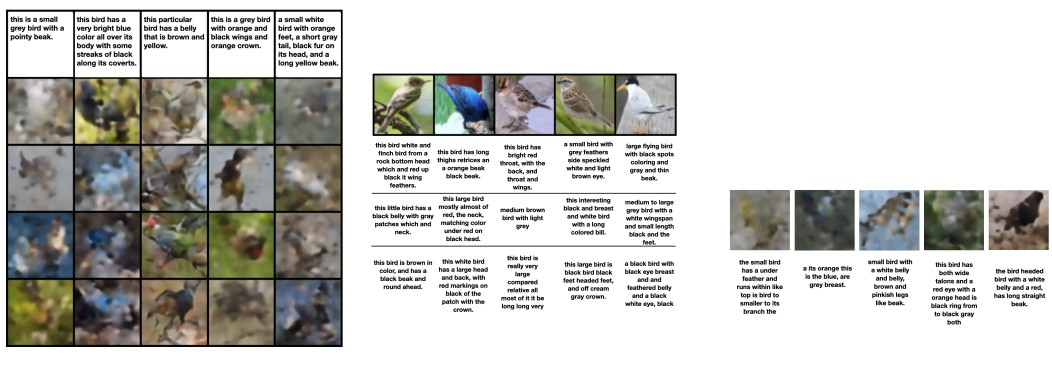

Figure 17: Qualitative results for DMVAE on the CUB Image-Captions experiment.

## H  PERFORMANCE OF MMVAE+ AS A FUNCTION OF THE NUMBER OF MODALITIES USED FOR TRAINING

An important trait of successful multimodal learning is that performance should improve, or at least non-decrease, when additional modalities are available. In fact, adding modalities increases, or at least does not decrease, the information available for the model. However, Daunhawer et al. (2022) highlight that for the prominent approaches in the class of multimodal VAEs having additional modalities causes a decrease in generative quality. Here we investigate the impact of varying the number of modalities present for training on the MMVAE+ performance. In particular we use the PolyMNIST dataset for an ablation of conditional generative coherence and conditional generative quality for MMVAE+, when varying the number of modalities for training. Note the PolyMNIST dataset consists of five modalities: $\{\mathbf{m}_0, \mathbf{m}_1, \mathbf{m}_2, \mathbf{m}_3, \mathbf{m}_4\}$. In this ablation, we compare MMVAE+ trained on

- five (all) modalities $\{\mathbf{m}_0, \mathbf{m}_1, \mathbf{m}_2, \mathbf{m}_3, \mathbf{m}_4\}$
- four modalities $\{\mathbf{m}_1, \mathbf{m}_2, \mathbf{m}_3, \mathbf{m}_4\}$
- three modalities $\{\mathbf{m}_2, \mathbf{m}_3, \mathbf{m}_4\}$
- two modalities $\{\mathbf{m}_3, \mathbf{m}_4\}$

Then, for a transparent comparison, we compute FID scores and generative coherences calculated for conditional generation for starting modality $\mathbf{m}_3$ and target modality $\mathbf{m}_4$, and vice-versa, averaging the results. This way we have a fair and unbiased comparison of cross-generation performance by only considering the last two modalities even in models trained with three/four/five modalities.

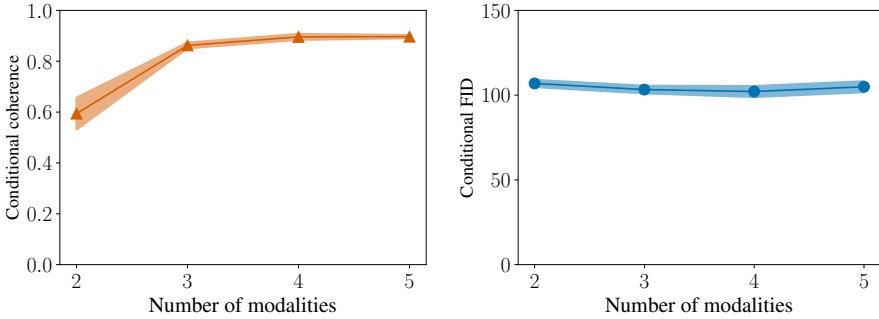

Figure 18: MMVAE+ performance for conditional generation on the PolyMNIST dataset, when varying the number of modalities used for training. $\beta = 2.5$ for all models.

Results suggest that we do not observe a decrease in generative quality when increasing the number of modalities, as it is observed for the main existing models in the class of multimodal VAEs (Daunhawer et al., 2022). On the other hand, generative coherence improves markedly with more modalities present for training. This result is important and promising, as it suggests that MMVAE+ can leverage multiple modalities to learn shared information effectively, while at the same time not incur in a trade-off with generative quality. In other words, having additional modalities seem to only bring a benefit, as it is desirable for multimodal VAEs, but currently not achieved for state-of-the-art approaches (Daunhawer et al., 2022).

## I  COMPARING MODELS WITH SEPARATE LATENT SUBSPACES USING DROPOUT DURING TRAINING

In this Appendix, we want to compare the MMVAE+ with existing models with factorized subspaces (see section 4.2), when dropout is applied to such models during training. In fact, Wang et al. (2016) suggest dropout as a potential solution for the shortcut problem. To this end we add dropout when training MMVAE(factorized), mmJSD(factorized), and DMVAE on PolyMNIST with the same settings as in Section 4.2, with equal shared and private latent capacity (setting that corresponds to

100% in the x-axis, in plots in Figure 5). Note that, with this choice for the number of dimensions for shared and modality-specific subspaces, all three models trained without dropout clearly show the presence of a shortcut (see again Figure 5). We use different dropout rates: 0.1, 0.2, 0.4; the same values proposed in Wang et al. (2016). To the comparison, we also add VCCA-private, the model proposed by the authors in Wang et al. (2016). Note that for this comparison, we extend the VCCA-private model to $M$ modalities and optimize the objective

$$\mathcal{L}_{\text{VCCA-private}}(\boldsymbol{x}_{1:M}) = \frac{1}{M} \sum_{m=1}^{M} \mathbb{E}_{\substack{q_{\phi_{\boldsymbol{z}_m}}(\boldsymbol{z}|\boldsymbol{x}_m) \\ q_{\Phi_{\boldsymbol{W}}}(\boldsymbol{W}|\boldsymbol{X})}} \left[ \log \frac{p_{\Theta}(\boldsymbol{X}, \boldsymbol{z}, \boldsymbol{W})}{q_{\phi_{\boldsymbol{z}_m}}(\boldsymbol{z}|\boldsymbol{x}_m) q_{\Phi_{\boldsymbol{W}}}(\boldsymbol{W}|\boldsymbol{X})} \right]$$

where $\boldsymbol{W} = \boldsymbol{w}_1, \dots \boldsymbol{w}_M$, $p_{\Theta}(\boldsymbol{X}, \boldsymbol{z}, \boldsymbol{W}) = p(\boldsymbol{z}) \prod_{m=1}^{M} p_{\theta_m}(\boldsymbol{x}_m|\boldsymbol{z}, \boldsymbol{w}_m) p(\boldsymbol{w}_m)$, and $q_{\Phi_{\boldsymbol{W}}}(\boldsymbol{W}|\boldsymbol{X}) = \prod_{m=1}^{M} q_{\phi_{\boldsymbol{w}_m}}(\boldsymbol{w}_m|\boldsymbol{x}_m)$.

For the compared models, we report in Figure 19 the performance for generative coherence for conditional generation, latent classification accuracy for private subspaces, and generative quality for conditional generation. Dropout rate 0.0 equals to models trained without dropout. In each plot, we keep as reference the performance of MMVAE+ trained without dropout, and with the same values for shared and private dimensionalities as all other models. Results show that increasing the dropout rate for existing approaches has a similar effect to constraining the size of private subspaces.

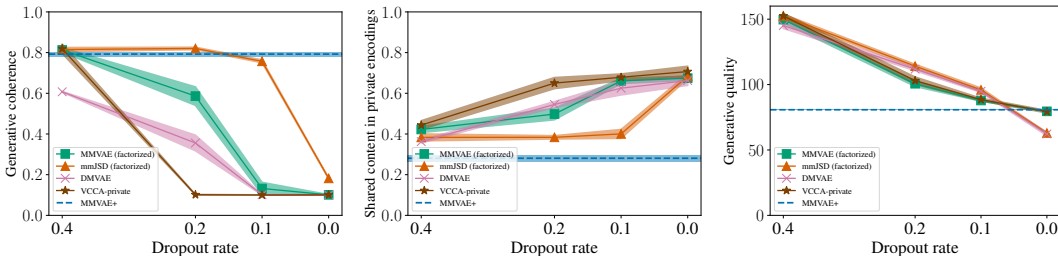

(a) Generative coherence (higher is better)

(b) Latent classification accuracy from modality-specific encodings (lower is better)

(c) Generative quality (in terms of FID score, lower is better)

Figure 19: Comparison of MMVAE+, MMVAE(factorized), mmJSD(factorized), DMVAE, VCCA-private, trained on PolyMNIST with equal number of dimensions assigned to shared and modality-specific subspaces: both the shared and the modality-specific subspaces are always fixed to 32 dimensions. MMVAE(factorized), mmJSD(factorized), DMVAE, VCCA-private are trained with different dropout rates, while MMVAE+ trained without dropout is kept as reference. Note we show dropout rates in decreasing order in the x-axis, to highlight the similarity in trend with the plots from Section 4.2: this shows increasing the dropout rate has a similar effect to constraining the size of private subspaces.

In particular increasing the dropout rate corresponds to improved generative coherence, and better results for latent classification accuracy for private subspaces. However, this comes at the expense of a markedly worsened generative quality. Therefore, increasing the dropout rate seems to prevent the degenerate solution in which all information flows in the modality-specific subspaces. However, it also results in low generative quality, which means it is not an effective solution to the shortcut problem, when looking at overall performance. This demonstrates that our novel solution of using auxiliary distributions is a more effective method to solve the shortcut problem. From these results it is evident that, even if compared with existing models resorting to dropout during training, MMVAE+ achieves better results when jointly looking at both generative quality and generative coherence.

