# OpenReview forum: "MMVAE+: Enhancing the Generative Quality of Multimodal VAEs without Compromises"
_ICLR.cc/2023/Conference — ICLR 2023 poster_

### Official Review · Reviewer_GWdS · 2022-10-16

**Confidence:** 5
**Correctness:** 3
**Technical Novelty And Significance:** 3
**Empirical Novelty And Significance:** 3
**Recommendation:** 8

**Clarity, Quality, Novelty And Reproducibility:**

The paper reads very well, it is clear that it presents high quality work and a thorough experimental campaign.

On the novelty, one could comment that the presented method is a variant (eq 5) of known methods, and that the key observation of trying to avoid “pollution” of private latent spaces is not new. However, I think the derivation to be correct, and — contrary to some previous work — the authors spent effort and energy in proving that the proposed ELBO is indeed a valid ELBO. In a previous version of this work, appeared as a workshop paper in ICLR 2022, the appendix show the proofs for Lemma 2 which I found interesting, because it attempts at showing that the parameters of the private encoders do not depend on cross-modal reconstruction. I wonder why in this version that lemma has been omitted.

Concerning the reproducibility of the work, I think this is somehow easy, as the key idea consists in a somehow “innocent” modification of the ELBO of the MMVAE approach. As such, this is a good point for this work.

**Strength And Weaknesses:**

Strength:

* This work follows a line of research that tackle an important problem, that of multimodal representation learning

* The paper is well written, clear, and easy to follow, and builds on many previous work, with the added value of clarifying some aspects of prior work that where not completely clear in the original papers

* Experiments are well conceived

Weaknesses:

* The idea presented in this work (to the best of my understanding) can be summarized as follows: shared information should not pollute modality-specific latent information. To ensure this, a variant of the mixture of expert ELBO is derived, by introducing learnable pseudo-priors that induce the reconstruction of a given modality to rely solely on its modality-specific latent space. A natural question is as follows: since you assume independence between shared and modality-specific latent spaces, and further you constrain modality-specific information to be solely responsible for reconstruction, then why not training individual VAEs, one per modality, then a multimodal VAE to learn a shared representation, and then “stitch” together all latent spaces?
In other words, what I miss is a naive baseline to double check that the idea presented in this paper is a valid approach when compared to a simpler method.

* I find Expression (5) in the main paper not very clear. In a previous workshop paper that presents the very same idea and the very same method I find the use of a tilde on the “other-modalities” private latent variables easier to understand and more clear. On the good side, I think the textual explanations of the ELBO are much more useful in this version of the paper.

* In the experiments, unconditional coherence seem to be a more difficult objective to attain, as was also shown in prior works. In this case, we usually sample from the prior on the latent space, and only rely on decoders to generate each modality. Then we check the coherence of each generated modality. 1) It would be useful to clarify how does this work when you have a latent space that is segregated. I assume you take the prior of each modality-specific latent, and the shared prior, but it would be good to state this clearly (in the appendix). Why do you only have results on PolyMNIST for this challenging metric, and not for CUB?
Moreover, when we look at table 3 in the appendix (for PolyMNIST), we can see that the proposed method really struggles to achieve high generation quality (there, the MVAE approach based on PoE works better). It is also interesting to note that additional “tricks” to improve base models (importance sampling, double reparametrization), seem to hurt generation quality, which is not expected.

* In the experiments, when we look at “competitors”, we learn that if the capacity is sufficient, then the modality-specific encoder can learn all the information (both shared and private) such that an adverse effect of a shortcut appears. But if all information is stored in the modality-specific latent space, and this is true for each modality, then wouldn’t this be enough to achieve coherence? Aren’t we simply looking at the “fallacy” of the MoE approach that, by construction, samples only one of the components for the generation?

* Fig 5 raise several questions.
** Fig 5b indicates latent classification accuracy, obtained by training a simple linear model on “freezed” encoders that produce latent features. It seems counter intuitive to state that lower accuracy is better. This is because the goal is to show that if we only use modality-specific encodings we should not be able to classify correctly, if the hypothesis of no shared information being present in those latent spaces. Beside the fact that I am not convinced that this metric (first introduced in Shi et al ‘19) is useful to state anything about the latent space natural segregation, I think that comments to this figure should be more informative.
** Fig 5 c: I assume this figure is for the conditional generation right? Since there are works that criticize the use of FID score to assess the generative quality [1], I wonder why we do not have a table with likelihood values, or Bit per Dimension, which can also be useful to assess the quality of the model.
** Overall, from the three sub-figures in fig 5, we can remark that mmJSD is a close competitor to MMVAE+. I think this should be acknowledged and commented.

* The last paragraph before section 4.3 claims for an optimal balance between coherence and generation quality. How can you claim optimality?

* In section 4.3, qualitative results for the CUB dataset, in terms of coherence, use an original metric discussed in the appendix. This is an important detail that I think should be discussed at least a minimum in the main paper. Indeed, the proposed metric is somehow questionable and should be put upfront for the reader to properly understand. In my opinion, it is not clear why you could have not used the same technique used in Shi et al ‘19, by taking the learned representation of your ResNet, in a similar way as if you used a pre-trained network. The introduction of artificial captions and the count of correctly colored pixels suffers from the problems stated by the authors, e.g. counting only the background color. Using the top-2 most frequent pixel colors might work in some cases an not others, so I am not sure how much can we rely on this coherence metric. Also, as commented above, this is conditional coherence, which seems an easier task to achieve than unconditional coherence. Do you have any comments why, and why didn’t you show unconditional coherence metric fo CUB?


[1] @misc{FID_critics,
  url = {https://arxiv.org/abs/2203.06026},
  author = {Kynkäänniemi, Tuomas and Karras, Tero and Aittala, Miika and Aila, Timo and Lehtinen, Jaakko},
  title = {The Role of ImageNet Classes in Fréchet Inception Distance},
  publisher = {arXiv},
  year = {2022},
}


**Summary Of The Paper:**

The work presented in this paper addresses the important problem of multimodal representation learning. Following a large body of work in this domain, the proposed approach builds on an extension of a Variational Auto Encoder (VAE) such that it can encode and decode multiple modalities. The crux of the idea presented in this paper is to partition the latent space into modality-specific and shared latent variables, making sure that the modality-specific latent are not “polluted” by the share information. This is achieved through a modification of the ELBO that originates from the work of Shi et al. ‘19, that uses a mixture of experts approach to obtain the joint approximate posterior. By decomposing the ELBO following Shi et al. ‘19, they observe that for a given modality, the portion that pertains to all other modalities should be trained by assuming a learned “pseudo” prior for their latent variables.
Experimental results use mainly two datasets: PolyMNIST, which is a variant of MNIST including background images to simulate multiple modalities, and the CUB dataset, which contains two modalities: images and captions. In their experiments, the authors rely on standard metrics from the literature (or their reinterpretation) including generative coherence (both conditional and unconditional) and generative quality (measured through the lenses of the FID score). Experiments serve the purpose of illustrating that the proposed method strikes a good balance between the two objectives (coherence and quality), which is a pain point of previous work.

**Summary Of The Review:**

In summary, this work is well presented, well written and the motivation is clear. I think this line of work presents the important trade-off between coherence and generation quality that affects VAE-based method for multimodal representation learning, and this is commendable.
I have spotted some problems that can be summarized as: 1) given the key idea in this work, it would be possible to conceive a naive baseline to make sure that the claims presented in the paper are indeed correct, and this is missing, 2) Experiments seem not to be conclusive about the superiority of the proposed method with respect to state of the art. There cases in which this seems true, and others in which the “optimality” of the proposed solution cannot be verified.

Lastly, since this work is not completely novel, and the strength are not on the methodological contributions, I would advice the authors to improve the experimental section, by considering at least one additional dataset [2], or [3].

[2] Vasco M, Yin H, Melo FS, Paiva A. Leveraging hierarchy in multimodal generative models for effective cross-modality inference. Neural Netw. 2022 Feb;146:238-255. doi: 10.1016/j.neunet.2021.11.019. Epub 2021 Nov 24. PMID: 34906760.


[3] @misc{https://doi.org/10.48550/arxiv.2107.07502,
  url = {https://arxiv.org/abs/2107.07502},
  author = {Liang, Paul Pu and Lyu, Yiwei and Fan, Xiang and Wu, Zetian and Cheng, Yun and Wu, Jason and Chen, Leslie and Wu, Peter and Lee, Michelle A. and Zhu, Yuke and Salakhutdinov, Ruslan and Morency, Louis-Philippe},
  title = {MultiBench: Multiscale Benchmarks for Multimodal Representation Learning},
  publisher = {arXiv},
  year = {2021},
}


============================

Post rebuttal comments

============================

I've read the various discussions in the rebuttal, and find the new results compelling in verifying the merits of the proposed method. I will update my score from 5 to 8.

---

> ### Author Response · Authors · 2022-11-18
> **Reply to Reviewer GWdS (part 1)**
>
>
> > The idea presented in this work (to the best of my understanding) can be summarized as follows: shared information should not pollute modality-specific latent information. To ensure this, a variant of the mixture of expert ELBO is derived, by introducing learnable pseudo-priors that induce the reconstruction of a given modality to rely solely on its modality-specific latent space. A natural question is as follows: since you assume independence between shared and modality-specific latent spaces, and further you constrain modality-specific information to be solely responsible for reconstruction, then why not training individual VAEs, one per modality, then a multimodal VAE to learn a shared representation, and then “stitch” together all latent spaces? In other words, what I miss is a naive baseline to double check that the idea presented in this paper is a valid approach when compared to a simpler method.
>
>
>
> Thank your for raising this question. We see the value of being able to compare a method to a simple baseline to certify that the idea has merit when compared to a simpler approach. Therefore here we discuss the nature of such a baseline in our setting, and its significance, and show a comparison with our approach.
>
> We do not believe that training unimodal VAEs and a joint multimodal VAE, all separately, and then concatenating the latent spaces, can be a sensible baseline. This is because in such a setting we train separate decoders to get as input the unimodal private representation **or** the joint multimodal representation but **not** both. This means we could never have both shared and modality-specific inforation as input to the decoder for generation. Instead, we describe in the following what the most straightforward approach one can conceive in this setting might be in our opinion, which can be seen as a baseline. Here we aim to learn a multimodal VAE that can learn a joint shared representation, and single unimodal representsions containing **only** modality-specific information, then use both a unimodal and the joint representation for reconstruction/generation.  Then one can learn a latent-variable model with a shared latent variable $\textbf{z}$ denoting shared information between modalities, and $\textbf{W} = \textbf{w}\_1, \dots, \textbf{w}\_M$ latent variables denoting modality-specific information for each modality. Note that independency is assumed between latent variables $\textbf{z}, \textbf{w}\_1, \dots, \textbf{w}\_M$. In line with these assumptions, the resulting generative model factorizes as $p_\Theta(\textbf{X}, \textbf{z}, \textbf{W}) = p(\textbf{z})  \prod\_{m=1}^M p(\textbf{w}\_m) p\_{\theta_m}(\textbf{x}\_m| \textbf{z}, \textbf{w}\_m)$. As true posteriors are intractable, one approximates posterior inference with encoders $q\_{\Phi\_{\textbf{z}}}(\textbf{z}| \textbf{X}), q\_{\phi\\_{\textbf{w}\_1}}(\textbf{w}\_1 | \textbf{x}\_1), \dots q\_{\phi\_{\textbf{w}\_M}} (\textbf{w}\_M | \textbf{x}\_M)$, where the modality-specific encoders are unimodal encoders, while the encoders for the shared subspace takes as inputs all modalities.
> Then the derived ELBO is
>
>
> \begin{align}
> \mathcal{L}(\textbf{X}) & = \mathbb{E}\_{\substack{ q\_{\Phi\_{\textbf{z}}}(\textbf{z} |\textbf{X}) \\\\ q\_{\phi\_{\textbf{w}\_1}}(\textbf{w}\_1 |\textbf{x}\_1) \\\\ \vdots \\ q\_{\phi\_{\textbf{w}\_M}}(\textbf{w}\_M |\textbf{x}\_M) }}\biggl[ \log \frac{p\_\Theta(\textbf{X}, \textbf{z}, \textbf{W})}{q\_{\Phi\_{\textbf{z}}}(\textbf{z} | \textbf{X}) \prod\_{m=1}^Mq\_{\phi\_{\textbf{w}\_m}}(\textbf{w}\_m |\textbf{x}\_m)} \biggr]
> \end{align}
>
> However, without further assumptions on the joint encoder, such a formulation is not scalable to large number of modalities, and for this reason we assume some form of decomposition of the joint encoder in terms of unimodal encoders. Note that if we assume a mixture-of-experts encoder we recover the MMVAE objective with factorized latent representations (Equation (4)). This shows that this model, which we compare to in our work, already fills the role of the simplest method one can apply in our context, and serves as a baseline. However, for completeness, here we compute an additional baseline by assuming a product-of-experts joint encoder in the objective above. This essentially corresponds to  a naive extnsion of the  MVAE (Wu and Goodman, 2018) to  factorized latent subspaces. The updates comparisons with this baseline included, can be found in the Appendix G of the updated manuscript.
>
>
> Note that both baselines heavily suffer from the shortcut problem described in our work, as other existing approaches with factored latent subspaces. Moreover, they fail to achieve sufficient performance: generative quality and generative coherence are never **both** present at the same time. With this we show that simpler baseline methods are not successful for the problem we aim to solve in this paper.
>
> **[Note: this reply continues in a separate comment]**

---

> ### Author Response · Authors · 2022-11-18
> **Reply to Reviewer GWdS (part 2)**
>
>
> > I find Expression (5) in the main paper not very clear. In a previous workshop paper that presents the very same idea and the very same method I find the use of a tilde on the “other-modalities” private latent variables easier to understand and more clear. On the good side, I think the textual explanations of the ELBO are much more useful in this version of the paper.
>
> We believe Expression (5) to be formally correct. However, we understand that using a tilde would make it clearer that those variables are sampled from the auxiliary priors. Therefore, in the updated version of the manuscript, we use that notation.
>
>
> > In the experiments, unconditional coherence seem to be a more difficult objective to attain, as was also shown in prior works. In this case, we usually sample from the prior on the latent space, and only rely on decoders to generate each modality. Then we check the coherence of each generated modality. 1) It would be useful to clarify how does this work when you have a latent space that is segregated. I assume you take the prior of each modality-specific latent, and the shared prior, but it would be good to state this clearly (in the appendix).
>
> Your understanding on how unconditional generation works for MMVAE+, having separated latent subspaces, is correct. We sample the shared latent $\textbf{z}$ from its prior $p(\textbf{z})$, and the modality-specific latent $\textbf{w}\_m$ for modality $m$ from the respective prior $p(\textbf{w}\_m)$. Thanks for the suggestion of stating this clearly in the Appendix, we will implement it in the updated manuscript.
>
> >  Moreover, when we look at table 3 in the appendix (for PolyMNIST), we can see that the proposed method really struggles to achieve high generation quality (there, the MVAE approach based on PoE works better).
>
> Looking at PolyMNIST results, the MVAE indeed achieves the best results for (unconditional) generative quality. However, note that  coherence for MVAE extremely is low. This means that its overall performance is actually far from good, as generative coherence is a key component of performance for multimodal VAEs. We stress once more here that good performance for multimodal VAEs requires **both high generative quality and high generative coherence**, and while certain competitors might show superior performance compared to MMVAE+ in one of the two metrics in certain cases, MMVAE+ is the only method that consistenly achieves **both high generative quality and high generative coherence in conditional and unconditional generation**.
>
>
> >  It is also interesting to note that additional “tricks” to improve base models (importance sampling, double reparametrization), seem to hurt generation quality, which is not expected.
>
> We agree that the effect of such techniques on the resuts can be object of investigation for future work. It seems to slightly improve coherence at the expense of a slight reduction of generative quality. In this work it made sense to show results for MMVAE and MMVAE+ trained with such techniques to be consistent with previous work (Shi et al (2019)). However, one notes that the advantage of MMVAE+ over its competitors, when looking at the presence of both high generative quality and high generative coherence, is maintained irrespectively of these techniques.
>
> **[Note: this reply continues in a separate comment]**

---

> ### Author Response · Authors · 2022-11-18
> **Reply to Reviewer GWdS (part 3)**
>
>
> > In section 4.3, qualitative results for the CUB dataset, in terms of coherence, use an original metric discussed in the appendix. This is an important detail that I think should be discussed at least a minimum in the main paper. Indeed, the proposed metric is somehow questionable and should be put upfront for the reader to properly understand. In my opinion, it is not clear why you could have not used the same technique used in Shi et al ‘19, by taking the learned representation of your ResNet, in a similar way as if you used a pre-trained network. The introduction of artificial captions and the count of correctly colored pixels suffers from the problems stated by the authors, e.g. counting only the background color. Using the top-2 most frequent pixel colors might work in some cases an not others, so I am not sure how much can we rely on this coherence metric. Also, as commented above, this is conditional coherence, which seems an easier task to achieve than unconditional coherence. Do you have any comments why, and why didn’t you show unconditional coherence metric fo CUB?
>
> >   Why do you only have results on PolyMNIST for this challenging metric [unconditional coherence], and not for CUB?
>
>
> It was challenging to come up with a quantitative proxy for conditional coherence in the CUB Image-Captions experiment. **In fact, the metric proposed by Shi et al. (2019) for coherence in this dataset is effective only when dealing with the simpler version of this experiment that uses pretrained ResNet-features, rather than training on actual images**. Shi et. al (2019) train the model on ResNet embeddings rather than real images: for image generation the model actually generates an embedding, that is matched with the 'nearest' embedding among all embeddings of the images in the original data. Therefore what the model outputs in an image belonging to the test set. This 'trick' results in having high quality images as output, but no generation in pixel space happens, which makes the task by far easier. Their coherence metric relies on the fact that generated images are essentially taken from the dataset itself, and therefore come from the same distribution as the data. So, cross-generated images can be fed to the pre-trained Resnet used at training to compute a meaningful embedding to be compared with the embedding for the starting image. In our case setting instead, images are generated directly in pixel space, which means there would be a distribution shift between the generated images and the starting images (e.g. generated images are blurrier and naturally less detailed). With such a significant distribution shift, feeding generated images to the ResNet encoders trained on original data does not yield meaninful results, which invalids such a metric for our setting, as also highlighted in recent work (Daunhawer et al. 2022).
>
> Notably, we were still able to design a proxy for conditional coherence for CUB Image-Captions, which fairly reflects qualitative assessment. It is, to the best of our knowledge, the first proposed quantitative proxy for conditional coherence based on generated images in this challenging setting. However, we still recognize its limitations, in that it needs a set of starting captions designed so to have single annotations about color. For unconditional generation, we do not have such a control setting, and therefore the metric does not trivially transfer.
>
> Finally, we thank you for your input. We are more specific in the Appendix of the updated version of the paper describing the reasons for which the same technique use by Shi et al. (2019) is not applicable in our experimental setting. For reasons of spaces, these specifics need to be left to the Appendix. However, we mention the novelty of our metric and inapplicability of previously designed ones in the main text.
>
> > The last paragraph before section 4.3 claims for an optimal balance between coherence and generation quality. How can you claim optimality?
>
> Thank you for this question, which allows us to correct a phrasing, that we see how it can result confusing. Here, we used a misinterpretable phrasing: we did not mean that we achieve the best possible balance between the two performance criteria, as in the best balance that can be possibly achieved by any model in this case. We rather meant to say the following. The best performance for MMVAE+, when jointly  looking at generative quality and generative coherence, is better when the shared and modality-specific subspaces are of comparable capacities, rather than when the private subspaces are constrained in size. This is evidence for the fact that the MMVAE+ effectively exploits adiditonal modality-specific capacity without incurring in a shortcut. We correct this phrasing in the updated version of the manuscript.
>
>    **[Note: this reply continues in a separate comment]**

---

> ### Author Response · Authors · 2022-11-18
> **Reply to Reviewer GWdS (part 4)**
>
>
>
> > In the experiments, when we look at “competitors”, we learn that if the capacity is sufficient, then the modality-specific encoder can learn all the information (both shared and private) such that an adverse effect of a shortcut appears. But if all information is stored in the modality-specific latent space, and this is true for each modality, then wouldn’t this be enough to achieve coherence? Aren’t we simply looking at the “fallacy” of the MoE approach that, by construction, samples only one of the components for the generation?
>
> It is rather the opposite: if all information is stored in modality-specific latent spaces, one gets the lowest possible values for coherence, corresponding to random choice.  In fact, consider cross-modal generation from modality $m\_1$ to modalitty $m\_2$. For this we compute the encoding for the shared space from the starting sample $\textbf{x}\_{m\_1}$, as $\textbf{z} \sim q\_{\phi\_{\textbf{z}\_{m\_1}}}(\textbf{z} | \textbf{x}\_{m\_1})$. Then the modality-specific encoding for private information for the target modality is sampled from the corresponding prior  $\textbf{w}\_{m\_2} \sim p(\textbf{w}\_{m\_2})$. The cross-generated sample is then $\bar{\textbf{x}}\_{m_2} \sim p\_{\theta\_{m\_2}}(\textbf{x}\_{m\_2} | \textbf{z}, \textbf{w}\_{m\_2})$. Note that, for cross-modal generation, modality-specific information is **always** sampled from a prior, as the target sample from the target modality is not present due to the nature of the task itself. Therefore, to reach coherence, shared information has to be contained in $\textbf{z}$. In contrast, if all information for each modality is stored in modality-specific spaces, it would lead to the worst possible coherence performance, as modality-specific encodings are sampled at random for cross-generation. Note, that the same conclusion holds for unconditional generation. There, again considering two modalities for brevity, we sample $\textbf{z} \sim p(\textbf{z})$, $\textbf{w}\_{m\_1} \sim p(\textbf{w}\_{m\_1})$,  $\textbf{w}\_{m\_2} \sim p(\textbf{w}\_{m\_2})$ and then sample $\bar{\textbf{x}}\_{m\_1} \sim p\_{\theta\_{m\_1}}(\textbf{x}\_{m\_1} | \textbf{z}, \textbf{w}\_{m\_1})$ and $\bar{\textbf{x}}\_{m\_1} \sim p\_{\theta\_{m\_2}}(\textbf{x}\_{m\_2} | \textbf{z}, \textbf{w}\_{m\_2})$. Hence, if shared information is encoded in modality-specific subspaces, given modality-specific encodings are again sampled independently from different priors, we would get no coherence among generated samples.
>
> Finally, we want to point out that our argument is not related to having a MoE encoder. The same argument would hold with any choice of decompositiion for the joint encoder (PoE, MoPoE), when we assume separate modality-specific spaces in our model. In summary, in any case we have to avoid having shared information encoded in modality-specific spaces to achieve coherence, and here is where the fallacy of existing apporaches with factorized subspaces comes from, as shown in our work.
>
> > Fig 5 raise several questions. ** Fig 5b indicates latent classification accuracy, obtained by training a simple linear model on “freezed” encoders that produce latent features. It seems counter intuitive to state that lower accuracy is better. This is because the goal is to show that if we only use modality-specific encodings we should not be able to classify correctly, if the hypothesis of no shared information being present in those latent spaces. Beside the fact that I am not convinced that this metric (first introduced in Shi et al ‘19) is useful to state anything about the latent space natural segregation, I think that comments to this figure should be more informative. **
>
> Previous work (Shi et al. (2019)) associates high latent classification accuracy, with a linear classifier, with the fact that shared and private information is factorized into separate **implicit** subspaces, of the same shared latent space. We agree with you that this association is questionable in some sense. However, this is not the usage of latent classification accuracy we make in our work. **We are primarly interested in proving the shared information cannot be retrieved from the modality-specific subspaces, when training with the MMVAE+ objective, as it is not contained in these subspaces (Fig 5b).** For this purpose, we believe latent classification accuracy is actually very effective metric, given we use a non-linear classifier instead of a linear classifier as done in previous work, in order to be sure to capture all information. Thank you for this question. We comment in more detail latent classification accuracy in Fig 5b.
>
> **[Note: this reply continues in a separate comment.]**

---

> ### Author Response · Authors · 2022-11-18
> **Reply to Reviewer GWdS (part 5)**
>
>
> > ** Fig 5 c: I assume this figure is for the conditional generation right? Since there are works that criticize the use of FID score to assess the generative quality [1], I wonder why we do not have a table with likelihood values, or Bit per Dimension, which can also be useful to assess the quality of the model.
>
> It is for conditional generation indeed, and particularly meant to compare generative quality. We do not consider likelihood values as they actually evaluate cross-modal **reconstruction** performnce, and good cross-modal reconstruction performance does not in practice correlate with good cross-modal **generation** performance, especially when looking at generative quality. In fact note that likelihood scores for conditional generation are computed by taking the test set with paired samples, and then computing the likelihood of reconstrucing **a given target sample** from a given starting modality. However, note that for conditional generation no information on modality-specific features for the target modality is available. Therefore, always predicting the average value for private features would yield the highest likelihood values. However, this would result in average-looking samples with very low generative quality. From these arguments one can conlude that likelihood values are not as effective as a metric in the multimodal setting, as they are for unimodal generative models, and hence possibly why recent work on multimodal VAEs has also resorted to FID scores instead (Hwang et al, 2021).
>
> > Overall, from the three sub-figures in fig 5, we can remark that mmJSD is a close competitor to MMVAE+. I think this should be acknowledged and commented.
>
> We respectfully disagree with this statement. Generative quality is indeed comparable. However, in this ablation conditional coherence and latent classification accuracy for private encodings are the central metrics. In both of these metrics there is an evident significant gap between mmJSD and MMVAE+, from which it is clear that mmJSD heavily suffers from the shortcut problem.
>
> > In a previous version of this work, appeared as a workshop paper in ICLR 2022, the appendix show the proofs for Lemma 2 which I found interesting, because it attempts at showing that the parameters of the private encoders do not depend on cross-modal reconstruction. I wonder why in this version that lemma has been omitted.
>
> Thank you for this comment. It led to the decision to include that result in the Appendix of the updated version of the manuscript, as we see how it can be useful to the understanding of our proposed objective.
>
>
> >  I would advice the authors to improve the experimental section, by considering at least one additional dataset [...].
>
> We believe that the realistic version of the CUB Image-Captions experiment we use in this work, where models are trained on real images, makes our empirical results already strong. Our approach is indeed able to achieve good performance in this setting, which was found to be prohibitive for existing approaches in recent work (Daunhawer et al (2022)). However, we appreciate the suggestion, and reserve the possibility to apply our method to other experimental settings for future work.

---

### Official Review · Reviewer_VT76 · 2022-10-25

**Confidence:** 3
**Correctness:** 4
**Technical Novelty And Significance:** 3
**Empirical Novelty And Significance:** 3
**Recommendation:** 6

**Clarity, Quality, Novelty And Reproducibility:**

The paper is well written. The presentation is clear with literature cited appropriately.

**Strength And Weaknesses:**

Strengths:
- Previous work on multi-modal generative models have struggled to yield both generative coherence and generative quality. The suggested approach illustrates an empirical improvement for both criteria on challenging multimodal datasets.
- The suggested variational objective is interesting and new, as far as I am aware. It offers a different approach to include modality specific latent variables compared to previous work that appears more robust.

Weaknesses:
- It is not clear to me if the gap between the variational bound and the log-likelihood does increase with increasing modalities (as in Daunhawer et al., 2022)?
- Do additional modalities compromise the generative performance? It would be helpful to see how the generative/cross-coherence performance varies empirically, for instance in the PolyMNIST experiments with different numbers of modalities.

Actionable feedback:
- The proportionality following eq. (4) is not clear to me.


Comments:
- It appears that the auxiliary prior distributions could be quite general. Is there a specific reason why they have been chosen to be zero centered Gaussians? Does the tightness of the bound relative to the log-likelihood depend on the KL divergence between $p$ and $r$ or $q(w_m|x_m)$ and $r(w_m)$, and could I improve generative quality, for example, by using implicit densities for r(as I do not need to evaluate their density)?



**Summary Of The Paper:**

The paper introduces a new multi-modal VAE model which includes modality-specific latent variables, as well as shared latent variables with the latter encoded through a mixture-of-experts model.  The new variational bound allows to utilize private latent variables that do not hinder cross-modal coherence and generation, in contrast to previous work where this does not hold in a robust way. Empirical results against different multi-modal VAEs for different benchmark datasets illustrate that the approach improves generative coherence and generative quality.

----
Post-rebuttal comments:

Following the response from the authors and the other reviews, I keep my accept recommendation.
The authors have addressed my question as to how the (conditional) generative performance changes when additional modalities are available. The new experiments indicate that generative performance does not decrease, in contrast to some previous work.
The additional experiments to address comments from the other reviewers also improve the experimental validation of the suggested approach.

----

**Summary Of The Review:**

The paper suggests a novel variational bound for multi-modal VAEs that seems well motivated and appears to yield good empirical results.

---

> ### Author Response · Authors · 2022-11-18
> **Reply to Reviewer VT76 (part 1)**
>
>
> > Do additional modalities compromise the generative performance? It would be helpful to see how the generative/cross-coherence performance varies empirically, for instance in the PolyMNIST experiments with different numbers of modalities.
>
>
> Thank you for raising this question, as how the model behaves when modalities increase in number is important. In fact, in the presence of succesful multimodal learning having more modalities should yield benefit on performance rather than be detrimental. In contrast, Daunhawer et al. (2022) have shown that for the existing models in the class of multimodal VAEs adding modalities reduces generative quality.
>
> Following your suggestion, we show an ablation of generative coherence and generative quality for MMVAE+, when varying the number of modalities for training on the PolyMNIST dataset. Note the PolyMNIST dataset consists of five modalities we label as $\{\textbf{m}_0, \textbf{m}_1, \textbf{m}_2, \textbf{m}_3, \textbf{m}_4 \}$. For such an ablation, we compare MMVAE+ trained on
> - five (all) modalities $\{\textbf{m}_0, \textbf{m}_1, \textbf{m}_2, \textbf{m}_3, \textbf{m}_4 \}$
> - four modalities $\{\textbf{m}_1, \textbf{m}_2, \textbf{m}_3, \textbf{m}_4 \}$
> - three modalities $\{ \textbf{m}_2, \textbf{m}_3, \textbf{m}_4 \}$
> - two modalities $\{ \textbf{m}_3, \textbf{m}_4 \}$
>
> Then, for a transparent comparison, we compute FID scores and generative coherences calculated for conditional generation for starting modality $\textbf{m}_3$ and target modality  $\textbf{m}_4$, and vice-versa, averaging the results. This way we have a fair and unbiased comparison of cross-generation performance by only considering the last two modalities even in models trained with 3/4/5 modalities.
>
>
> Results, shown in Appendix H of the updated manuscript, suggest that we do not observe a decrease in generative quality when increasing the number of modalities, as it is demonstrated for the main existing models in the class of multimodal VAEs (Daunhawer et al., 2022). On the other hand, generative coherence improves markedly with more modalities present for training. This result is important and promising, as it shows that MMVAE+ can leverage multiple modalities to learn shared information effectively, while at the same time not incur in a trade-off with generative quality. In other words, having additional modalities seem to only bring a benefit, as it is desirable for multimodal VAEs, but currently not achieved for state-of-the-art approaches (Daunhawer et al., 2022).
>
> > It is not clear to me if the gap between the variational bound and the log-likelihood does increase with increasing modalities (as in Daunhawer et al., 2022)?
>
> We believe the derivation from Daunhawer et al. (2022) can still apply to our training objective, and thank you for this question which is a great input for future work. However, while theoretically significant, such a conclusion does not necessarily have a strong meaning for performance at test time. In other words, it is sometimes non-trivial to make the connection between higher likelihoods and better performance in the context of multimodal VAEs. As an exampe, having the highest cross-modal reconstruction likelihoods at test time, and therfore a higher test ELBO, would not yield good performance for cross-modal generation. In fact, for cross-modal reconstruction, no information on modality-specific features for the target modality is available, and therfore always predicting the average value for private features would yield the highest likelihood values. However, this would result in average-looking samples with very low generative quality, which is not to our interest at test time. Note that in fact, our empirical results suggest that performance, particularly for cross-modal generation, does not degrade when additional modalities are used for training.
>
> **[Note: this reply continues in a separate comment.]**

---

> ### Author Response · Authors · 2022-11-18
> **Reply to Reviewer VT76 (part 2)**
>
> > The proportionality following eq. (4) is not clear to me.
>
> We do not know if this is what raised the confusion, but we spotted a small typo in that expression. Thank you for making us notice. In any case, that is fixed now, and we show how the proportionality holds explicitly here
>
> \begin{align}\mathbb{E}\_{\substack{q\_{\phi\_{\textbf{z}\_m}}(\textbf{z} | \textbf{x}\_m)\\\\q\_{\Phi\_{\textbf{W}}}(\textbf{W} | \textbf{X})}}\biggl[\log\frac{p\_{\Theta}(\textbf{X}, \textbf{z}, \textbf{W})}{q\_{\Phi\_{\textbf{z}}}(\textbf{z}|\textbf{X}) q\_{\Phi\_{\textbf{W}}}(\textbf{W}|\textbf{X})}\biggr] & = \mathbb{E}\_{\substack{q\_{\phi\_{\textbf{z}\_m}}(\textbf{z} | \textbf{x}\_m)\\\\q\_{\Phi\_{\textbf{W}}}(\textbf{W}|\textbf{X})}}\biggl[\log\frac{ p(\textbf{z}) p(\textbf{W})\prod\_{m=1}^Mp\_{\theta\_m}(\textbf{x}\_m| \textbf{z}, \textbf{w}\_m)}{q\_{\Phi\_{\textbf{z}}}(\textbf{z}|\textbf{X}) q\_{\Phi\_{\textbf{W}}}(\textbf{W}|\textbf{X})}\biggr] \\\\& = \mathbb{E}\_{\substack{q\_{\phi\_{\textbf{z}\_m}}(\textbf{z} |\textbf{x}\_m)\\\\q\_{\phi\_{\textbf{w}\_m}}(\textbf{w}\_m| \textbf{x}\_m)}}\biggl[\log p\_{\theta\_{m}}(\textbf{x}\_m| \textbf{z}, \textbf{w}\_m)\biggr] + \sum^M\_{\substack{m=1 \\\\n\neq m}} \mathbb{E}\_{\substack{q\_{\phi\_{\textbf{z}\_m}}(\textbf{z} | \textbf{x}\_m)\\\\q\_{\phi\_{\textbf{w}\_n}}(\textbf{w}\_n | \textbf{x}\_n)}}\biggl[\log p\_{\theta\_{n}}(\textbf{x}\_n| \textbf{z}, \textbf{w}\_n)\biggr] \\\\ & + \mathbb{E}\_{\substack{q\_{\phi\_{\textbf{z}\_m}}(\textbf{z}| \textbf{x}\_m)\\\\q\_{\Phi\_{\textbf{W}}}(\textbf{W}| \textbf{X})}}\biggl[\log\frac{p(\textbf{z}) p(\textbf{W})}{q\_{\Phi\_{\textbf{z}}}(\textbf{z}|\textbf{X}) q\_{\Phi\_{\textbf{W}}}(\textbf{W}|\textbf{X})}\biggr] \\\\
> &  \propto \mathbb{E}\_{\substack{q\_{\phi\_{\textbf{z}\_m}}(\textbf{z}| \textbf{x}\_m)\\\\q\_{\phi\_{\textbf{w}\_m}}(\textbf{w}\_m | \textbf{x}\_m)}}\biggl[\log p\_{\theta\_{m}}(\textbf{x}\_m| \textbf{z}, \textbf{w}\_m)\biggr] + \sum^M\_{\substack{m=1 \\\\n\neq m}} \mathbb{E}\_{\substack{q\_{\phi\_{\textbf{z}\_m}}(\textbf{z} | \textbf{x}\_m )\\\\q\_{\phi\_{\textbf{w}\_n}}(\textbf{w}\_n | \textbf{x}\_n)}}\biggl[\log p\_{\theta\_{n}}(\textbf{x}\_n| \textbf{z}, \textbf{w}\_n)\biggr] \end{align}
>
>
> > It appears that the auxiliary prior distributions could be quite general. Is there a specific reason why they have been chosen to be zero centered Gaussians? Does the tightness of the bound relative to the log-likelihood depend on the KL divergence between $p$ and $r$ or $q(w_m | x_m)$ and $r(w_m)$ and could I improve generative quality, for example, by using implicit densities for r(as I do not need to evaluate their density)?
>
> It is indeed true, our derivations hold for an almost arbitrary choice of auxiliary prior distributions. It is certainly interesting to study how different choices could impact performance, and a great input for future work. The rationale behind the choice of zero-mean priors is that it seemed sensible to have $r(\textbf{w}_m)$ and $p(\textbf{w}_m)$ centered on the same value, but need not necessarily be the only sensible choice. We also did not test whether using implicit densities could yield an improvement. However, we discuss in depth in Appendix C why our choice of assuming  zero-mean priors with parameterized variance as auxiliary distributions is convenient and yields good performance results, while being a fairly general choice, that only inducts bias in the choice of the location parameter (set to zero). Finally, since the auxiliary priors do not appear in any KL-divergence term in the objective, nor in our derivations, our intuition would be that the tightness of the bound relative to the log-likelihood does not depend on the KL divergence between $p$ and $r$ or $q(w_m | x_m)$ and $r(w_m)$. However, we reserve the possibility to provide a formal proof for future work.

---

> > ### Comment · Reviewer_VT76 · 2022-11-30
> > **Minor comment regarding the ‘proportionality’**
> >
> > Is this meant to mean equal up to an additive constant that depends only on the decoder parameters?

---

> > > ### Author Response · Authors · 2022-12-02
> > > **Re: Minor comment regarding the ‘proportionality’**
> > >
> > > Thank you for this question, and we hope with this comment we clarify the matter.
> > >
> > > We would like to highlight that the main purpose of the expression following Equation (4) in the paper is to make explicit how  the computation of the term on the left-hand side subsumes the computation of the two summands on the right-hand side, which is formally true, as it can be seen from our previous [comment](https://openreview.net/forum?id=sdQGxouELX&noteId=NL31N4b0w-). The rationale behind this is making the self-reconstruction and cross-reconstruction terms explicit in the objective in (4), to the eyes of the reader. In fact, in the paragraph that follows in the paper we discuss how self-reconstructions and cross-reconstructions are computed in this objective.
> > >
> > > After your valuable feedback, we thought of directly removing the proportionality and putting the following expression below equation (4), in the camera-ready version of our paper.
> > >
> > > \begin{align}\mathbb{E}\_{\substack{q\_{\phi\_{\textbf{z}\_m}}(\textbf{z} | \textbf{x}\_m)\\\\q\_{\Phi\_{\textbf{W}}}(\textbf{W} | \textbf{X})}}\biggl[\log\frac{p\_{\Theta}(\textbf{X}, \textbf{z}, \textbf{W})}{q\_{\Phi\_{\textbf{z}}}(\textbf{z}|\textbf{X}) q\_{\Phi\_{\textbf{W}}}(\textbf{W}|\textbf{X})}\biggr] & = \mathbb{E}\_{\substack{q\_{\phi_{\textbf{z}\_m}}(\textbf{z} |\textbf{x}\_m)\\\\q\_{\phi\_{\textbf{w}\_m}}(\textbf{w}\_m| \textbf{x}\_m)}}\biggl[\log p\_{\theta\_{m}}(\textbf{x}\_m| \textbf{z}, \textbf{w}\_m)\biggr] + \sum^M\_{\substack{n=1 \\\\n\neq m}} \mathbb{E}\_{\substack{q\_{\phi\_{\textbf{z}\_m}}(\textbf{z} | \textbf{x}\_m)\\\\q\_{\phi\_{\textbf{w}\_n}}(\textbf{w}\_n | \textbf{x}\_n)}}\biggl[\log p\_{\theta\_{n}}(\textbf{x}\_n| \textbf{z}, \textbf{w}\_n)\biggr]  + \mathbb{E}\_{\substack{q\_{\phi\_{\textbf{z}\_m}}(\textbf{z}| \textbf{x}\_m)\\\\q\_{\Phi\_{\textbf{W}}}(\textbf{W}| \textbf{X})}}\biggl[\log\frac{p(\textbf{z}) p(\textbf{W})}{q\_{\Phi\_{\textbf{z}}}(\textbf{z}|\textbf{X}) q\_{\Phi\_{\textbf{W}}}(\textbf{W}|\textbf{X})}\biggr]
> > > \end{align}
> > > It is the straightforward decomposition of the left-hand side in three summands (same as in our calculations in the previous [comment](https://openreview.net/forum?id=sdQGxouELX&noteId=NL31N4b0w-)). This expression already  makes explicit the self-reconstruction (first summand) and cross-reconstruction (second summand) terms in the objective, which are central to the comments on this objective that follow in the paper, so it serves to our purpose. At the same time, it would not raise any unnecessary confusion that can stem from resorting to the proportionality.
> > >
> > > We hope with this comment we address your concern, and we thank you for raising this point which can result in improved clarity in this part of our paper. Also, we hope the insights we provided in the rebuttal following your thoughtful questions addressed your previous concerns, maybe even towards a possible adjustment of the score.

---

### Official Review · Reviewer_i3y6 · 2022-10-25

**Confidence:** 3
**Correctness:** 4
**Technical Novelty And Significance:** 2
**Empirical Novelty And Significance:** 2
**Recommendation:** 6

**Clarity, Quality, Novelty And Reproducibility:**

The paper is written with good clarity in terms of how the ideas and model formulations are conveyed to the readers.

Extensive experiments justify the problems the authors raise and the effectiveness of the solution they provided.

Although with good experiments and results, the paper lacks contribution and novelty to the problem and the solution, as factorizing subspaces for private and shared features of different modalities and introducing cross-modal reconstruction loss have been implemented in (Lee, Pavlovic 2021) paper, which should also be included as a baseline.  Comparing Eq (4) with (5),  (5) is adding the sum(n!=m) term in (4) again, with the auxiliary distributions. It would be good to have an ablation study for the auxiliary distributions to show the novelty.


**Strength And Weaknesses:**

Strengths：
The paper is well-written, clear and well-organized.
Experiments show their model achieves both good generative coherence and high generative quality and is robust to the size of the private space.

Weaknesses:
(Lee, Pavlovic 2021) should also be included as a baseline.
It would be good to have an ablation study for the auxiliary distributions.
The figures can be improved:
It would be better to have an explanation on the boxes, lines and dashed lines for Figure 2.
It would be better to have an explanation for the other rows in Figure 3, why do they come in blocks?
The axis should be consistent in Figure 5. In (b),  lower is better but 0 is at the top, which confuses the readers.


**Summary Of The Paper:**

This paper provides a modified version of the existing MMVAE model called MMVAE+ for weakly-supervised generative learning with multiple modalities. The paper aims to overcome the trade-off between generative quality and generative coherence by suggesting having separate latent encoding for both shared and private features within each modality and a new ELBO. The authors also introduce auxiliary distributions for private features to achieve the cross-modal reconstruction. Experiments show their model achieves both good generative coherence and high generative quality in challenging experiments.


**Summary Of The Review:**

Although the paper is well written and the arguments are well justified, the paper lacks technical and empirical novelty to contribute as the problem and solution has been raised by an existing publication.

-----------------------------------------------------------------------
Post rebuttal comments:

After reading the rebuttal and the other reviews, we raise the score to 6.

As one of the contributions is the auxiliary distribution, it would be interesting to see how different choices of auxiliary distributions would affect the experiment results.

---

> ### Author Response · Authors · 2022-11-18
> **Reply to Reviewer i3y6 (part 1)**
>
> > Weaknesses: (Lee, Pavlovic 2021) should also be included as a baseline.
>
> > Although with good experiments and results, the paper lacks contribution and novelty to the problem and the solution, as factorizing subspaces for private and shared features of different modalities and introducing cross-modal reconstruction loss have been implemented in (Lee, Pavlovic 2021) paper, which should also be included as a baseline.
>
> Thank you for suggesting a comparison with this DMVAE (Lee, Pavlovic 2021), as we believe a comparison with this approach further shows the importance of our contribution in uncovering the shortcut problem for existing models with factorized subspaces and proposing our approach as a solution. We introduce DMVAE as a baseline in our comparison in Section 4.2, where we compare our approach with existing models with separate subspaces. We show that the DMVAE heavily suffers from the shortcut problem, as the other existing approaches with factorized subspaces.
>
> In detail, Lee and Pavlovic (2021) propose the DMVAE: a product-of-experts objective, with the addition of cross-modal reconstruction and unimodal reconstruction terms, to learn factorizing subspaces for private and shared features. In the following, we show how our model clearly separates from the DMVAE from both a theoretical and an empirical standpoint. **First and foremost, compared to our approach, the DMVAE objective is not provably an ELBO.** This is an important theoretical difference with MMVAE+, and calls for questioning whether DMVAE can be fully considered as a member of the class of multimodal VAEs. Note that in the comparisons we included in our submission, we restricted to models that provably optimize a valid ELBO. However, we now updated the comparisons for Section 4.2 in the updated version of the manuscript to include DMVAE. These results show that, like other existing approaches with factorized subspaces, **DMVAE heavily suffers from the shortcut problem we uncover in our work**, and is markedly unstable with respect to hyperparameters. **Our approach, on the contrary shows high robustness with respect to changes in hyperparameters controlling private latent dimensionality.** This shows that the shortcut problem we uncover and solve in this work, is not limited to mixture-based approaches but rather common to existing models that assume factorizes latent subspaces. This consolidates the generality and validity of our contribution.
>
> In light of these findings, it is safe to argue that there are both methodological and crucial empirical differences between our work and the work from Lee and Pavlovic (2021). We believe both (Sutter et al. (2020)) and (Lee and Pavlovic (2021)) have merit in contributing to the direction of designing multimodal VAEs with factorized latent subspaces. However, our work takes a significant step further by uncovering important limitations of these approaches, and proposing a novel approach that clearly separates from the existing ones, and results in crucial performance improvements and increased applicability to real-world datasets.
>
>
> Note we also test DMVAE on the CUB Image-Captions experiment for completeness, and show these results in the Appendix. Results show that the model clearly underperforms in this experiment with respect to MMVAE+.
>
>
> > It would be good to have an ablation study for the auxiliary distributions.
> > Comparing Eq (4) with (5), (5) is adding the sum(n!=m) term in (4) again, with the auxiliary distributions. It would be good to have an ablation study for the auxiliary distributions to show the novelty.
>
> We believe an ablation that shows the novelty and validity of our approach, of which an integral component is the usage of auxiliary distributions, is already in our work. In fact, we introduced the factorized version of the MMVAE objective (to the best of our knowledge, not explitly introduced in previous work) as an intermediate step between the MMVAE and the MMVAE+. The difference between the factorized version of the MMVAE objective and the MMVAE+ is the usage of auxiliary distrbutions during training. Therefore, our direct comparisons between the performance achieved with the MMVAE (factorized) objective and the MMVAE+ objective shown in Section 4.2, already shows the novelty and significance of our idea of using auxiliary distributions.
>
> >  The figures can be improved: It would be better to have an explanation on the boxes, lines and dashed lines for Figure 2
>
> Thank you very much for the input. We  add the details of the used notation for Figure 2 in the updated version of the manuscript.
>
> **[Note: this reply continues in a separate comment]**

---

> > ### Comment · Reviewer_i3y6 · 2022-11-30
> > **Clarification on auxiliary distribution ablation**
> >
> > If I understand correctly,
> >
> > 1. The idea of shared and private spaces is not novel, as it already appeared in  (Wang et a. (2016)), (Sutter et al. (2020)) and (Lee and Pavlovic (2021)).
> >
> > 2. There are two differences between eq (4) and eq(5):
> > a. the $\prod_{n\neq m}p_{\theta_n}(x_n|z,\tilde{w_n})$ part inside the log, but that looks like adding $\sum_{m=1,n\neq m}^ME(\log p_{\theta_n}(x_n|z,\tilde{w_n}))$ in eq(4) again. Can this be viewed as a contribution? And how does it solve the shortcut problem?
> >
> > b. the auxiliary distributions.
> >
> > Can the contributions caused by both a and b be separated (i.e. ablation on the auxiliary distributions)?

---

> > > ### Author Response · Authors · 2022-12-01
> > > **Re: Clarification on auxiliary distribution ablation**
> > >
> > >
> > > Thank you for your comment.
> > >
> > > > The idea of shared and private spaces is not novel, as it already appeared in (Wang et a. (2016)), (Sutter et al. (2020)) and (Lee and Pavlovic (2021)).
> > >
> > > We already acknoledged these previous works in our paper (Related work, last paragraph), and discussed their relevance. In both our paper (Section 4.2), and [in the scope of this rebuttal](https://openreview.net/forum?id=sdQGxouELX&noteId=P5lDL4XhIU), we already provided in-depth insights demostrating how having separate shared and private subspaces alone is not sufficient for good and robust performance of multimodal VAEs, even when resorting to training strategies such as dropout. We already, both in our paper and in the scope of this rebuttal, provided insights on the relevant weaknesses of existing models with separate shared and private subspaces, which our model overcomes (Section 4.2 and this [comment](https://openreview.net/forum?id=sdQGxouELX&noteId=P5lDL4XhIU)).
> > >
> > > >There are two differences between eq (4) and eq(5): a. the $\prod\_{n \neq m} p\_{\theta\_n}(x\_n | z, \tilde{w\}_n)$ part inside the log, but that looks like adding $\sum^M\_{m=1, n \neq m} E(\log p\_{\theta_n}(x\_n | z, \tilde{w}\_n))$ in eq(4) again. Can this be viewed as a contribution? And how does it solve the shortcut problem?
> > > b. the auxiliary distributions.
> > > Can the contributions caused by both a and b be separated (i.e. ablation on the auxiliary distributions)?
> > >
> > > In our previous [comment](https://openreview.net/forum?id=sdQGxouELX&noteId=M_mQo37ggr), we already explain why we introduce the objective in (4) in our work, and make it clear that the difference with the MMVAE+ objective is the usage of auxiliary distiributions to compute cross-reconstructions. We have demonstrated the MMVAE+ to be a novel objective, and an important contribution, with extensive empirical results in our paper and in the scope of this rebuttal.
> > >
> > > Regarding your comment that in Equation (5) "looks like we add $\sum^M\_{m=1, n \neq m} E(\log p\_{\theta_n}(x\_n | z, \tilde{w}\_n))$ again" with respect to equation (4), we would like to clarify that it is not the case that we add an additional term  to equation (4). Instead, the correct view is that cross-modal reconstructions are computed differently in the two objectives. The objectives do not differ the way you describe in a. This is clear from Equations (4) and (5), as well as explicit if looking at the derivation in Appendix A. The b. point on the other hand, i.e. auxiliary distributions, is the crucial difference between the objectives, as we already explained in our previous [comment](https://openreview.net/forum?id=sdQGxouELX&noteId=M_mQo37ggr). Therefore, the results you ask for boil down to an ablation proving the value of the novelty of using auxiliary distrbutions.
> > >
> > > As we stated already in our previous [comment](https://openreview.net/forum?id=sdQGxouELX&noteId=M_mQo37ggr), the ablation for auxiliary distributions you request, an ablation that shows the value of using auxiliary distributions in the MMVAE+ objective, is already present in our work (Section 4.2). In particular the ablation is the comparison between MMVAE(factorized), model trained with objective (4), and MMVAE+, model trained with objective (5), part of the results contained in Section 4.2. In fact, the crucial difference between the objective we introduce in (4) and the MMVAE+ objective we introduce in (5) is that the MMVAE+ ELBO makes use of auxiliary distributions for cross-modal reconstruction. Additional comparisons with other existing models with separate subspaces, made even more extensive [in the scope of this rebuttal](https://openreview.net/forum?id=sdQGxouELX&noteId=P5lDL4XhIU), further prove the relevance of our novel method in the field of multimodal generative models.
> > >
> > > We are glad you asked for clarifications, and hope this comment clarifies the answers for your questions, that we had already addressed in our previous comments. We also have acknowledged and addressed your main concern from the review, i.e. a comparison with the work of Lee and Pavlovic (2021), in our previous [comment](https://openreview.net/forum?id=sdQGxouELX&noteId=M_mQo37ggr). A direct comparison with this method, as well as with other existing methods with separate subspaces, has clarified the relevance of our contribution. We even show more insights for that comparison, to be found in our [answer](https://openreview.net/forum?id=sdQGxouELX&noteId=P5lDL4XhIU) to the comment by the  AC. If you should have any additional clarifications to ask, please do not hesitate to do so. Otherwise, we hope that we having addressed your concerns results in an adjustment of the score.

---

> > > > ### Comment · Reviewer_i3y6 · 2022-12-01
> > > > **Updated official review**
> > > >
> > > > Thanks for your reply. We have updated the official review.

---

> > > > > ### Author Response · Authors · 2022-12-04
> > > > > **Thank you for your feedback!**
> > > > >
> > > > > We are glad to see our reply resolved your concerns, and thank you for your feedback.

---

> ### Author Response · Authors · 2022-11-18
> **Reply to Reviewer i3y6 (part 2)**
>
> > It would be better to have an explanation for the other rows in Figure 3, why do they come in blocks?
>
> If the question refers to the black square that surrounds MMVAE and MMVAE+ results, we put it just to highlight the direct comparison between the two models. Thank you for the input, and we remove it in the updated version of the manuscript so not to raise any confusion. If instead you're seeking an explanation for what different rows in the image mean, we have the first row showcasing the starting samples from the first modality, which are used for conditional generation to each of the remaining modalities. The remaining sixteen rows below showcase four instances of conditionally generated samples with the second, third, fourth and fifth modality as target modality, in this order.
>
> > The axis should be consistent in Figure 5. In (b), lower is better but 0 is at the top, which confuses the readers.
>
> We made this design choice of the figure as to have the best performance always at the top of each plot, for an easier comparison between models to the reader's eyes. However, we see how this can actually cause some confusion given we then state how 'lower is better'. In the updated version of the manuscript we update the figure to always have the conventional increasing order for absolute values in the y-axis.

---

### Author Response · Authors · 2022-11-18
**Reply to all reviewers**

Dear reviewers,

we are grateful to you all for the constructive feedback.

We appreciate reviewers recognizing the relevance of the problem we tackle ( **VT76**,**GWdS**), the quality of the conceived experimental setting (**i3y6**, **VT76**,**GWdS**),and the strength of our presented results (**i3y6**,**VT76**). Also, we are glad all reviewers appraise the clarity of our exposition.

In this rebuttal, we address the concerns and questions that have been raised by the reviewers. In detail:
- Reviewer **i3y6**: in response to your concern on the significance of our contribution, when in comparison with the model presented in Lee, Pavlovic (2021), we make an extensive comparison between our approach and the one presented in Lee, Pavlovic (2021) in the scope of this rebuttal, showing a clear separation both from a conceptual and an empirical standpoint. Furthermore, we include the given model as a baseline in our updated version of the manuscript. Thank you for your suggestion, as we believe including Lee, Pavlovic (2021) in our comparisons further elucidates how our model improves over existing work. In addition we further elaborate on the role of auxiliary priors, clarifying where in our work we already show results that serve as a study on the role of this component of our model.
- Reviewer **VT76**: we sincerely thank you for pointing our attention to the performance of our model when varying the number of modalities.  We address your questions with both empirical results and discussions, gaining important insights on the value of our contribution compared to previous works.
- Reviewer **GWdS**: we recognize the value of showing the relevance of our idea when compared to a simpler method, seen as a baseline. We discuss two baseline methods for the problem we tackle in our work, with a direct comparison with our approach, consolidating the importance and significance of our contribution. We highly value all the useful and thoughtful comments and questions you posed, and address them in the scope of this rebuttal.

For convenience, we put here the bibliography for the cited works throughout our replies.

- Imant Daunhawer, Thomas M. Sutter, Kieran Chin-Cheong, Emanuele Palumbo, and Julia E Vogt. On the limitations of multimodal VAEs. *In International Conference on Learning Representations*, 2022.
- HyeongJoo Hwang, Geon-Hyeong Kim, Seunghoon Hong, and Kee-Eung Kim. Multi-view representation learning via total correlation objective. *In Advances in Neural Information Processing Systems*, 2021.
- Mihee Lee and Vladimir Pavlovic. Private-shared disentangled multimodal vae for learning of latent representations. *In CVPR Workshop on Multimodal Learning and Applications*, 2021.
- Yuge Shi, N. Siddharth, Brooks Paige, and Philip Torr. Variational mixture-of-experts autoencoders for multi-modal deep generative models. *In Advances in Neural Information Processing Systems*, 2019.
- Thomas M. Sutter, Imant Daunhawer, and Julia E. Vogt. Multimodal generative learning utilizing jensen-shannon-divergence. In *Advances in Neural Information Processing Systems*, 2020
- Mike Wu and Noah Goodman. Multimodal generative models for scalable weakly-supervised learning. *In Advances in Neural Information Processing Systems*, 2018.

---

> ### Comment · Reviewer_GWdS · 2022-11-21
> **Thank you for your detailed responses**
>
> Dear authors,
>
> thank you for your detailed responses, that are clear and provide evidence of the merits of your proposed approach. The modifications for the revised paper you discuss are reasonable and will hopefully lead to an improved paper.
>
> I am leaning toward increasing the score I gave to the submission, but will wait for the end of the discussion phase to implement my changes.
>
> Thanks for your work.

---

### Comment · Area_Chair_S37n · 2022-11-21
**Missing reference**

Dear authors,

In my opinion, the paper and even some of the references miss relevant literature in the direction of multi-modal VAE:

Wang et al. Deep Variational Canonical Correlation Analysis. 2016.

IIUC, this reference already achieved the objective function (4) for two views and realized that the reconstruction shortcut exists due to private variables for each view, and discovered that dropout was effective in preventing shortcuts.

If possible, I would be happy to hear the authors opinion on the comparison between the proposed paper and Wang et al, either with theoretical arguments or empirical results.

Best,
AC

---

> ### Author Response · Authors · 2022-11-26
> **Reply to AC (part 1)**
>
> Thank you for showing interest in our work, and for this comment, which brings to our attention the work from Wang et al (2016), which we were not familiar with. This previous work proposes a deep variational approach for multi-view data with separate shared and private latent subspaces. In particular, the authors realize how having private variables for each view can lead to a degenerate solution (what we call shortcut in our work) where the shared subspaces is ignored. The authors then propose dropout as a solution to prevent shortcuts.
>
> In this reply, we compare our work with Wang et al (2016) from both a theoretical and empirical standpoint.
>
> **Theoretical comparison**
>
> > IIUC, this reference already achieved the objective function (4) for two views
>
> We believe you refer to the bi-VCCA-private objective in Wang et al (2016). This objective is indeed similar to the MMVAE objective (Shi et. at ,2019), extented to factorized subspaces (Equation (4) in our work). The MMVAE objective is not limited to two modalities, while the objective from Wang et al (2016) is limited to two modalities in the original work. However, the  bi-VCCA-private objective can be easily extended to multiple modalities, and compared with Equation (4).
> In particular, with the assumptions on the generative model stated in Section 3.2 in our work, one derives the bi-VCCA-private (Wang et al., 2016) objective for $M$ modalities
> \begin{align}
> \mathcal{L}\_{VCCA}(\textbf{X}) = \frac{1}{M}\sum\_{m=1}^M
> \mathbb{E}\_{\substack{q\_{\phi\_{\textbf{z}\_m}}(\textbf{z} | \textbf{x}\_m)\\\\q\_{\Phi\_{\textbf{W}}}(\textbf{W} | \textbf{X})}}\biggl[\log\frac{p\_{\Theta}(\textbf{X}, \textbf{z}, \textbf{W})}{{\color{red}{q\_{\phi\_{\textbf{z}\_m}}(\textbf{z}|\textbf{x}\_m)}}q\_{\Phi\_{\textbf{W}}}(\textbf{W}|\textbf{X})}\biggr]
> \end{align}
> which can be compared with Equation (4)
> \begin{align}
> \mathcal{L}^{f}\_{MMVAE}(\textbf{X}) = \frac{1}{M}\sum\_{m=1}^M
> \mathbb{E}\_{\substack{q\_{\phi\_{\textbf{z}\_m}}(\textbf{z} | \textbf{x}_m)\\\\q\_{\Phi\_{\textbf{W}}}(\textbf{W} | \textbf{X})}}\biggl[\log\frac{p\_{\Theta}(\textbf{X}, \textbf{z}, \textbf{W})}{{\color{red}{q\_{\Phi\_{\textbf{z}}}(\textbf{z}|\textbf{X})}} q\_{\Phi\_{\textbf{W}}}(\textbf{W}|\textbf{X})}\biggr]
> \end{align}
> The objectives are indeed similar, but there is a crucial difference. In fact, note that in the MMVAE objective the joint (mixture-of-experts) posterior for $\textbf{z}$ appears in the denominator of the fraction, while in the VCCA objective the unimodal posterior for $\textbf{z}$ appears instead. We highlight this difference in the formulas with red color. This difference is relevant from a theoretical and conceptual standpoint, as the MMVAE assumes a joint posterior over the shared latent subspace $\textbf{z}$, which decomposes as a mixture-of-experts of unimodal encoders. Instead the VCCA objective cannot be traced back to a specific choice of joint posterior: in fact a joint posterior is not even assumed. Instead, an average of ELBOs is maximized, where for each ELBO the shared latent code is inferred from a different modality with the corresponding unimodal encoder.
>
>
> We have analysed differences between the objective from Wang et al. (2016) and Equation (4).
>
>
> Our proposed model, the MMVAE+ uses the novel idea of auxiliary distributions. To establish the validity of our contribution, we now compare the MMVAE+ to previous approaches, including the model from Wang et al. (2016), when dropout is used during training. In fact Wang et al. (2016) propose this strategy to avoid a shortcut when using private subspaces.
>
> **[Note: this reply continues in a separate comment.]**

---

> ### Author Response · Authors · 2022-11-26
> **Reply to AC (part 2)**
>
>
> **Empirical comparison**
>
> Here we want to compare the MMVAE+ with existing models with factorized subspaces (see section 4.2), when dropout is applied to such models. In fact, Wang et al. (2016) suggest dropout as a solution to the shortcut problem. To this end we add dropout when training MMVAE(factorized), mmJSD(factorized), DMVAE on PolyMNIST with the same settings as in Section 4.2, with equal shared and private latent capacity (settings corresponding to 100% in the x-axis, in plots in Figure 5). Note that, with this choice for the number of dimensions for shared and modality-specific subspaces, all three models trained without dropout clearly show the presence of a shortcut (see Figure 5). We use different dropout rates: 0.1, 0.2, 0.4; the same values proposed in Wang et al. (2016). **To the comparison, we also add the model proposed by Wang et al. (2016) (Equation above), denoted as VCCA . This way we show a direct comparison between the model from Wang et al (2016), also when trained with dropout as suggested by the authors, and our model.** We report the performance  for generative coherence for conditional generation, latent classification accuracy for private subspaces, and generative quality for conditional generation (see link below). Dropout rate 0.0 equals to models trained without dropout. In each plot, we keep as reference the performance of MMVAE+ **trained without dropout**, and with the same values for shared and private dimensionalities as all other models. Note we show dropout rates in decreasing order in the x-axis, to highlight the similarity in trend with the plots from Section 4.2 in our work: this shows increasing the dropout rate has a similar effect to constraining the size of private subspaces.
>
> https://ibb.co/26WPDbk
>
> Results show that increasing the dropout rate for existing approaches has a similar effect to constraining the size of private subspaces. In particular increasing the dropout rate corresponds to improved generative coherence, and better results for latent classification accuracy for private subspaces. However, this comes at the expense of a markedly worsened generative quality. Therefore, **increasing the dropout rate seems to prevent the degenerate solution in which all information flows in the modality-specific subspaces. However, it also results in very low generative quality, which means it is not an effective solution to the shortcut problem, when looking at overall performance.** This demonstrates that our novel solution of using auxiliary distributions is a more effective method to solve the shortcut problem. From these results it is evident that, even if  compared with existing models that resort to dropout during training, MMVAE+ is the only model achieving **both** high generative quality and high generative coherence.
>
>
>
> In light of these considerations, we will include the work from Wang et al. (2016) in the camera-ready version of our paper, and mention their contribution in recognizing the shortcut problem. We will also mention, possibly in the Appendix, their proposed solution of using dropout, which is sensible for preventing the shared subspace from being ignored, but has detrimental effects on generative quality, and therefore is not an effective solution when looking at overall model performance. Instead we propose an effective solution in our work. Our model once again proves to be the only method achieving **both** high generative quality and high generative coherence robustly with respect to shared and private latent dimensionality, even if compared to existing approaches using specific training strategies such as dropout. In summary, we believe that our analysis and the new results clarify our contribution and its novelty and value in the field of multimodal generative models.
>
> We hope this answer addresses your concerns. If you have additional questions or would like any clarification, please do not hesitate to ask.
>
>
> - Weiran Wang, Xinchen Yan, Honglak Lee, and Karen Livescu. Deep Variational Canonical Correlation Analysis. arXiv preprint arXiv:1610.03454., 2016.

---

> > ### Comment · Reviewer_GWdS · 2022-11-29
> > **Very interesting**
> >
> > Dear authors,
> > thank you for the additional material, discussions and insights, following the lead by the AC.
> >
> > I think the contributions of the proposed model are clear, and I am satisfied with your rebuttal. I will revise my score accordingly.

---

> > > ### Author Response · Authors · 2022-11-30
> > > **Thank you for your feedback!**
> > >
> > > We highly appreciate your careful consideration of the rebuttal and the follow-up discussions. We are very glad to see that our answers addressed your concerns, and thank you for your feedback.

---

### Decision · Program_Chairs · 2023-01-20

**Decision:**

Accept: poster

**Justification For Why Not Higher Score:**

While the solution for avoiding the shortcut is smart, after all it is a relatively small modification of the baseline approach.

**Justification For Why Not Lower Score:**

The paper proposes a simple yet effective solution to the challenging issue of learning private and shared variables for multi-view data. It makes a contribution to the literature.

**Metareview: Summary, Strengths And Weaknesses:**

The paper introduces a new multi-modal VAE model which includes modality-specific latent variables, as well as shared latent variables with the latter encoded through a mixture-of-experts model. The new variational bound allows to utilize private latent variables that do not hinder cross-modal coherence and generation, and at the same time avoid shortcuts for generation. Empirical results against different multi-modal VAEs for different benchmark datasets illustrate that the approach improves generative coherence and generative quality.

Strength:
The method is well-motivated and the derivation/results are convincing. The authors also addressed the review comments well.

Weakness:
There are some references missing by the paper (a common issue for this specific field), and more detailed comparisons with missing references are desired.

**Note From Pc:**

if the above contains the word "oral" or "spotlight" please see: "oral" presentation means -> notable-top-5% and "spotlight" means -> notable-top-25%. As stated in our emails, we are disassociating presentation type from AC recommendations